# Doubly Accelerated
# Stochastic Variance Reduced Dual Averaging Method for Regularized Empirical Risk Minimization

**Tomoya Murata**

NTT DATA Mathematical Systems Inc. , Tokyo, Japan

`murata@msi.co.jp`

**Taiji Suzuki**

Department of Mathematical Informatics
Graduate School of Information Science and Technology, The University of Tokyo, Tokyo, Japan
PRESTO, Japan Science and Technology Agency, Japan
Center for Advanced Integrated Intelligence Research, RIKEN, Tokyo, Japan

`taiji@mist.i.u-tokyo.ac.jp`

## Abstract

We develop a new accelerated stochastic gradient method for efficiently solving the convex regularized empirical risk minimization problem in mini-batch settings. The use of mini-batches has become a golden standard in the machine learning community, because the mini-batch techniques stabilize the gradient estimate and can easily make good use of parallel computing. The core of our proposed method is the incorporation of our new "double acceleration" technique and variance reduction technique. We theoretically analyze our proposed method and show that our method much improves the mini-batch efficiencies of previous accelerated stochastic methods, and essentially only needs size $\sqrt{n}$ mini-batches for achieving the optimal iteration complexities for both non-strongly and strongly convex objectives, where $n$ is the training set size. Further, we show that even in non-mini-batch settings, our method achieves the best known convergence rate for non-strongly convex and strongly convex objectives.

## 1 Introduction

We consider a composite convex optimization problem associated with regularized empirical risk minimization, which often arises in machine learning. In particular, our goal is to minimize the sum of finite smooth convex functions and a relatively simple (possibly) non-differentiable convex function by using first order methods in mini-batch settings. The use of mini-batches is now a golden standard in the machine learning community, because it is generally more efficient to execute matrix-vector multiplications over a mini-batch than an equivalent amount of vector-vector ones each over a single instance; and more importantly, mini-batch techniques can easily make good use of parallel computing.

Traditional and effective methods for solving the abovementioned problem are the "proximal gradient" (PG) method and "accelerated proximal gradient" (APG) method [10, 3, 20]. These methods are well known to achieve linear convergence for strongly convex objectives. Particularly, APG achieves optimal iteration complexities for both non-strongly and strongly convex objectives. However, these methods need a per iteration cost of $O(nd)$, where $n$ denotes the number of components of the finite sum, and $d$ is the dimension of the solution space. In typical machine learning tasks, $n$ and $d$

correspond to the number of instances and features respectively, which can be very large. Then, the per iteration cost of these methods can be considerably high.

A popular alternative is the "stochastic gradient descent" (SGD) method [19, 5, 17]. As the per iteration cost of SGD is only $O(d)$ in non-mini-batch settings, SGD is suitable for many machine learning tasks. However, SGD only achieves sublinear rates and is ultimately slower than PG and APG.

Recently, a number of stochastic gradient methods have been proposed; they use a variance reduction technique that utilizes the finite sum structure of the problem ("stochastic averaged gradient" (SAG) method [15, 16], "stochastic variance reduced gradient" (SVRG) method [6, 22] and SAGA [4]). Even though the per iteration costs of these methods are same as that of SGD, they achieve a linear convergence for strongly convex objectives. Consequently, these methods dramatically improve the total computational cost of PG. However, in size $b$ mini-batch settings, the rate is essentially $b$ times worse than in non-mini-batch settings (the extreme situation is $b = n$ which corresponds to PG). This means that there is little benefit in applying mini-batch scheme to these methods.

More recently, several authors have proposed accelerated stochastic methods for the composite finite sum problem ("accelerated stochastic dual coordinate ascent" (ASDCA) method [18], Universal Catalyst (UC) [8], "accelerated proximal coordinate gradient" (APCG) method [9], "stochastic primal-dual coordinate" (SPDC) method [23], and Katyusha [1]). ASDCA (UC), APCG, SPDC and Katyusha essentially achieve the optimal total computational cost[1] for strongly convex objectives[2] in non-mini-batch settings. However, in size $b$ mini-batch settings, the rate is essentially $\sqrt{b}$ times worse than that in non-mini-batch settings, and these methods need size $O(n)$ mini-batches for achieving the optimal iteration complexity[3], which is essentially the same as APG. In addition, [12, 13] has proposed the "accelerated mini-batch proximal stochastic variance reduced gradient" (AccProxSVRG) method and its variant, the "accelerated efficient mini-batch stochastic variance reduced gradient" (AMSVRG) method. In non-mini-batch settings, AccProxSVRG only achieves the same rate as SVRG. However, in mini-batch settings, AccProxSVRG significantly improves the mini-batch efficiency of non-accelerated variance reduction methods, and surprisingly, AccProxSVRG essentially only needs size $O(\sqrt{\kappa})$ mini-batches for achieving the optimal iteration complexity for strongly convex objectives, where $\kappa$ is the condition number of the problem. However, the necessary size of mini-batches depends on the condition number and gradually increases when the condition number increases and ultimately matches with $O(n)$ for a large condition number.

**Main contribution**

We propose a new accelerated stochastic variance reduction method that achieves better convergence than existing methods do, and it particularly takes advantages of mini-batch settings well; it is called the "doubly accelerated stochastic variance reduced dual averaging" (DASVRDA) method. We describe the main feature of our proposed method below and list the comparisons of our method with several preceding methods in Table 1.

> Our method significantly improves the mini-batch efficiencies of state-of-the-art methods. As a result, our method essentially only needs size $O(\sqrt{n})$ mini-batches[4] for achieving the optimal iteration complexities for both non-strongly and strongly convex objectives.

Table 1: Comparisons of our method with SVRG (SVRG$^{++}$ [2]), ASDCA (UC), APCG, SPDC, Katyusha and AccProxSVRG. $n$ is the number of components of the finite sum, $d$ is the dimension of the solution space, $b$ is the mini-batch size, $L$ is the smoothness parameter of the finite sum, $\mu$ is the strong convexity parameter of objectives, and $\varepsilon$ is accuracy. "Necessary mini-batch size" indicates the order of the necessary size of mini-batches for achieving the optimal iteration complexities $O(\sqrt{L/\mu}\log(1/\varepsilon))$ and $O(\sqrt{L/\varepsilon})$ for strongly and non-strongly convex objectives, respectively. We regard one computation of a full gradient as $n/b$ iterations in size $b$ mini-batch settings, for a fair comparison. "Unattainable" implies that the algorithm cannot achieve the optimal iteration complexity even if it uses size $n$ mini-batches. $\widetilde{O}$ hides extra log-factors.

| | $\mu$-strongly convex | | | Non-strongly convex | | |
|---|---|---|---|---|---|---|
| | Total computational cost in size $b$ mini-batch settings | Necessary mini-batch size $L/\mu \geq n$ | otherwise | Total computational cost in size $b$ mini-batch settings | Necessary mini-batch size $L/\varepsilon \geq n\log^2(1/\varepsilon)$ | otherwise |
| SVRG $(^{++})$ | $O\left(d\left(n + \frac{bL}{\mu}\right)\log\left(\frac{1}{\varepsilon}\right)\right)$ | Unattainable | Unattainable | $O\left(d\left(n\log\left(\frac{1}{\varepsilon}\right) + \frac{bL}{\varepsilon}\right)\right)$ | Unattainable | Unattainable |
| ASDCA (UC) | $\widetilde{O}\left(d\left(n + \sqrt{\frac{nbL}{\mu}}\right)\log\left(\frac{1}{\varepsilon}\right)\right)$ | Unattainable | Unattainable | $\widetilde{O}\left(d\left(\frac{n+\sqrt{nbL}}{\sqrt{\varepsilon}}\right)\right)$ | Unattainable | Unattainable |
| APCG | $O\left(d\left(n + \sqrt{\frac{nbL}{\mu}}\right)\log\left(\frac{1}{\varepsilon}\right)\right)$ | $O(n)$ | $O(n)$ | No direct analysis | Unattainable | Unattainable |
| SPDC | $O\left(d\left(n + \sqrt{\frac{nbL}{\mu}}\right)\log\left(\frac{1}{\varepsilon}\right)\right)$ | $O(n)$ | $O(n)$ | No direct analysis | Unattainable | Unattainable |
| Katyusha | $O\left(d\left(n + \sqrt{\frac{nbL}{\mu}}\right)\log\left(\frac{1}{\varepsilon}\right)\right)$ | $O(n)$ | $O(n)$ | $O\left(d\left(n\log\left(\frac{1}{\varepsilon}\right) + \sqrt{\frac{nbL}{\varepsilon}}\right)\right)$ | $O(n)$ | $O(n)$ |
| AccProxSVRG | $O\left(d\left(n + \left(\frac{n-b}{n-1}\right)\frac{L}{\mu} + b\sqrt{\frac{L}{\mu}}\right)\log\left(\frac{1}{\varepsilon}\right)\right)$ | $O\left(\sqrt{\frac{L}{\mu}}\right)$ | $O\left(n\sqrt{\frac{\mu}{L}}\right)$ | No direct analysis | Unattainable | Unattainable |
| **DASVRDA** | $O\left(d\left(n + (b+\sqrt{n})\sqrt{\frac{L}{\mu}}\right)\log\left(\frac{1}{\varepsilon}\right)\right)$ | $O(\sqrt{n})$ | $O\left(n\sqrt{\frac{\mu}{L}}\right)$ | $O\left(d\left(n\log\left(\frac{1}{\varepsilon}\right) + (b+\sqrt{n})\sqrt{\frac{L}{\varepsilon}}\right)\right)$ | $O(\sqrt{n})$ | $\widetilde{O}\left(n\sqrt{\frac{\varepsilon}{L}}\right)$ |

## 2 Preliminary

In this section, we formally describe the problem to be considered in this paper and the assumptions for our theory.

We use $\|\cdot\|$ to denote the Euclidean $L_2$ norm $\|\cdot\|_2$: $\|x\| = \|x\|_2 = \sqrt{\sum_i x_i^2}$. For natural number $n$, $[n]$ denotes set $\{1, \ldots, n\}$.

In this paper, we consider the following composite convex minimization problem:

$$\min_{x\in\mathbb{R}^d} \{P(x) \stackrel{\text{def}}{=} F(x) + R(x)\}, \tag{1}$$

where $F(x) = \frac{1}{n}\sum_{i=1}^n f_i(x)$. Here each $f_i : \mathbb{R}^d \to \mathbb{R}$ is a $L_i$-smooth convex function and $R : \mathbb{R}^d \to \mathbb{R}$ is a relatively simple and (possibly) non-differentiable convex function. Problems of this form often arise in machine learning and fall under regularized empirical risk minimization (ERM). In ERM problems, we are given $n$ training examples $\{(a_i, b_i)\}_{i=1}^n$, where each $a_i \in \mathbb{R}^d$ is the feature vector of example $i$, and each $b_i \in \mathbb{R}$ is the label of example $i$. Important examples of ERM in our setting include linear regression and logistic regression with Elastic Net regularizer $R(x) = \lambda_1\|\cdot\|_1 + (\lambda_2/2)\|\cdot\|_2^2$ ($\lambda_1, \lambda_2 \geq 0$).

We make the following assumptions for our analysis:

**Assumption 1.** There exists a minimizer $x_*$ of (1).

**Assumption 2.** Each $f_i$ is convex, and is $L_i$-smooth, i.e.,

$$\|\nabla f_i(x) - \nabla f_i(y)\| \leq L_i\|x - y\| \ \ (\forall x, y \in \mathbb{R}^d).$$

**Assumption 3.** Regularization function $R$ is convex, and is relatively simple, which means that computing the proximal mapping of $R$ at $y$, $\mathrm{prox}_R(y) = \mathrm{argmin}_{x\in\mathbb{R}^d}\left\{\frac{1}{2}\|x - y\|^2 + R(x)\right\}$, takes $O(d)$ computational cost, for any $y \in \mathbb{R}^d$.

We always consider Assumptions 1, 2 and 3 in this paper.

**Assumption 4.** There exists $\mu > 0$ such that objective function $P$ is $\mu$-optimally strongly convex, i.e., $P$ has a minimizer and satisfies

$$\frac{\mu}{2}\|x - x_*\|^2 \leq P(x) - P(x_*) \ \ (\forall x \in \mathbb{R}^d, \forall x_* \in \mathrm{argmin}_{x\in\mathbb{R}^d} f(x)).$$

Note that the requirement of optimally strong convexity is weaker than the one of ordinary strong convexity (for the definition of ordinary strong convexity, see [11]).

We further consider Assumption 4 when we deal with strongly convex objectives.

# 3 Our Approach: Double Acceleration

In this section, we provide high-level ideas of our main contribution called "double acceleration."

First, we consider deterministic PG (Algorithm 1) and (non-mini-batch) SVRG (Algorithm 2). PG is an extension of the steepest descent to proximal settings. SVRG is a stochastic gradient method using the variance reduction technique, which utilizes the finite sum structure of the problem, and it achieves a faster convergence rate than PG does. As SVRG (Algorithm 2) matches with PG (Algorithm 1) when the number of inner iterations $m$ equals 1, SVRG can be seen as a generalization of PG. The key element of SVRG is employing a simple but powerful technique called the variance reduction technique for gradient estimate. The variance reduction of the gradient is realized by setting $g_k = \nabla f_{i_k}(x_{k-1}) - \nabla f_{i_k}(\widetilde{x}) + \nabla F(\widetilde{x})$ rather than vanilla stochastic gradient $\nabla f_{i_k}(x_{k-1})$. Generally, stochastic gradient $\nabla f_{i_k}(x_{k-1})$ is an unbiased estimator of $\nabla F(x_{k-1})$, but it may have high variance. In contrast, $g_k$ is also unbiased, and one can show that its variance is "reduced"; that is, the variance converges to zero as $x_{k-1}$ and $\widetilde{x}$ to $x_*$.

---

**Algorithm 1:** PG $(\widetilde{x}_0, \eta, S)$

  **for** $s = 1$ to $S$ **do**
    $\widetilde{x}_s =$ One Stage PG$(\widetilde{x}_{s-1}, \eta)$.
  **end for**
  **return** $\frac{1}{S} \sum_{s=1}^{S} \widetilde{x}_s$.

---

**Algorithm 2:** SVRG $(\widetilde{x}_0, \eta, m, S)$

  **for** $s = 1$ to $S$ **do**
    $\widetilde{x}_s =$ One Stage SVRG$(\widetilde{x}_{s-1}, \eta, m)$.
  **end for**
  **return** $\frac{1}{S} \sum_{s=1}^{S} \widetilde{x}_s$.

---

**Algorithm 3:** One Stage PG $(\widetilde{x}, \eta)$

  $\widetilde{x}^+ = \text{prox}_{\eta R}(\widetilde{x} - \eta \nabla F(\widetilde{x}))$.
  **return** $\widetilde{x}^+$.

---

**Algorithm 4:** One Stage SVRG $(\widetilde{x}, \eta, m)$

  $x_0 = \widetilde{x}$.
  **for** $k = 1$ to $m$ **do**
    Pick $i_k \in [1, n]$ randomly.
    $g_k = \nabla f_{i_k}(x_{k-1}) - \nabla f_{i_k}(\widetilde{x}) + \nabla F(\widetilde{x})$.
    $x_k = \text{prox}_{\eta R}(x_{k-1} - \eta g_k)$.
  **end for**
  **return** $\frac{1}{n} \sum_{k=1}^{n} x_k$.

---

**Algorithm 5:** APG $(\widetilde{x}_0, \eta, S)$

  $\widetilde{x}_{-1} = \widetilde{x}_0, \widetilde{\theta}_0 = 0$.
  **for** $s = 1$ to $S$ **do**
    $\widetilde{\theta}_s = \frac{s+1}{2}, \quad \widetilde{y}_s = \widetilde{x}_{s-1} + \frac{\widetilde{\theta}_{s-1}-1}{\widetilde{\theta}_s}(\widetilde{x}_{s-1} - \widetilde{x}_{s-2})$.
    $\widetilde{x}_s =$ One Stage PG$(\widetilde{y}_s, \eta)$.
  **end for**
  **return** $x_S$.

---

Next, we explain the method of accelerating SVRG and obtaining an even faster convergence rate based on our new but quite natural idea "outer acceleration." First, we would like to remind you that the procedure of deterministic APG is given as described in Algorithm 5. APG uses the famous "momentum" scheme and achieves the optimal iteration complexity. Our natural idea is replacing One Stage PG in Algorithm 5 with One Stage SVRG. With slight modifications, we can show that this algorithm improves the rates of PG, SVRG and APG, and is optimal. We call this new algorithm outerly accelerated SVRG. However, this algorithm has poor mini-batch efficiency, because in size $b$ mini-batch settings, the rate of this algorithm is essentially $\sqrt{b}$ times worse than that of non-mini-batch settings. State-of-the-art methods APCG, SPDC, and Katyusha also suffer from the same problem in the mini-batch setting.

Now, we illustrate that for improving the mini-batch efficiency, using the "inner acceleration" technique is beneficial. The author of [12] has proposed AccProxSVRG in mini-batch settings. AccProxSVRG applies the momentum scheme to One Stage SVRG, and we call this technique "inner" acceleration. He showed that the inner acceleration could significantly improve the mini-batch efficiency of vanilla SVRG. This fact indicates that inner acceleration is essential to fully utilize the mini-batch settings. However, AccProxSVRG is not a truly accelerated method, because in non-mini-batch settings, the rate of AccProxSVRG is same as that of vanilla SVRG.

In this way, we arrive at our main high-level idea called "double" acceleration, which involves applying momentum scheme to both outer and inner algorithms. This enables us not only to lead to

the optimal total computational cost in non-mini-batch settings, but also to improving the mini-batch efficiency of vanilla acceleration methods.

We have considered SVRG and its accelerations so far; however, we actually adopt stochastic variance reduced dual averaging (SVRDA) rather than SVRG itself, because we can construct lazy update rules of (innerly) accelerated SVRDA for sparse data. In Section G of supplementary material, we briefly discuss a SVRG version of our proposed method and provide its convergence analysis.

## 4 Algorithm Description

In this section, we describe the concrete procedure of the proposed algorithm in detail.

### 4.1 DASVRDA for non-strongly convex objectives

---

**Algorithm 6:** DASVRDA$^{\mathrm{ns}}(\widetilde{x}_0, \widetilde{z}_0, \gamma, \{L_i\}_{i=1}^n, m, b, S)$

$\widetilde{x}_{-1} = \widetilde{z}_0, \widetilde{\theta}_0 = 0, \bar{L} = \frac{1}{n}\sum_{i=1}^n L_i, Q = \{q_i\} = \left\{\frac{L_i}{n\bar{L}}\right\}, \eta = \frac{1}{\left(1 + \frac{\gamma(m+1)}{b}\right)\bar{L}}$.

**for** $s = 1$ to $S$ **do**

$\quad \widetilde{\theta}_s = \left(1 - \frac{1}{\gamma}\right)\frac{s+2}{2}, \quad \widetilde{y}_s = \widetilde{x}_{s-1} + \frac{\widetilde{\theta}_{s-1}-1}{\widetilde{\theta}_s}(\widetilde{x}_{s-1} - \widetilde{x}_{s-2}) + \frac{\widetilde{\theta}_{s-1}}{\widetilde{\theta}_s}(\widetilde{z}_{s-1} - \widetilde{x}_{s-1}).$

$\quad (\widetilde{x}_s, \widetilde{z}_s) = $ One Stage AccSVRDA$(\widetilde{y}_s, \widetilde{x}_{s-1}, \eta, m, b, Q).$

**end for**

**return** $\widetilde{x}_S$.

---

**Algorithm 7:** One Stage AccSVRDA $(\widetilde{y}, \widetilde{x}, \eta, m, b, Q)$

$x_0 = z_0 = \widetilde{y}, \bar{g}_0 = 0, \theta_0 = \frac{1}{2}$.

**for** $k = 1$ to $m$ **do**

$\quad$ Pick independently $i_k^1, \ldots, i_k^b \in [1, n]$ according to $Q$, set $I_k = \{i_k^\ell\}_{\ell=1}^b$.

$\quad \theta_k = \frac{k+1}{2}, \quad y_k = \left(1 - \frac{1}{\theta_k}\right)x_{k-1} + \frac{1}{\theta_k}z_{k-1}.$

$\quad g_k = \frac{1}{b}\sum_{i \in I_k}\frac{1}{nq_i}\left(\nabla f_i(y_k) - \nabla f_i(\widetilde{x})\right) + \nabla F(\widetilde{x}), \quad \bar{g}_k = \left(1 - \frac{1}{\theta_k}\right)\bar{g}_{k-1} + \frac{1}{\theta_k}g_k.$

$\quad z_k = \underset{z \in \mathbb{R}^d}{\mathrm{argmin}}\left\{\langle\bar{g}_k, z\rangle + R(z) + \frac{1}{2\eta\theta_k\theta_{k-1}}\|z - z_0\|^2\right\} = \mathrm{prox}_{\eta\theta_k\theta_{k-1}R}\left(z_0 - \eta\theta_k\theta_{k-1}\bar{g}_k\right).$

$\quad x_k = \left(1 - \frac{1}{\theta_k}\right)x_{k-1} + \frac{1}{\theta_k}z_k.$

**end for**

**return** $(x_m, z_m)$.

---

We provide details of the doubly accelerated SVRDA (DASVRDA) method for non-strongly convex objectives in Algorithm 6. Our momentum step is slightly different from that of vanilla deterministic accelerated methods: we not only add momentum term $((\widetilde{\theta}_{s-1} - 1)/\widetilde{\theta}_s)(\widetilde{x}_{s-1} - \widetilde{x}_{s-2})$ to the current solution $\widetilde{x}_{s-1}$ but also add term $(\widetilde{\theta}_{s-1}/\widetilde{\theta}_s)(\widetilde{z}_{s-1} - \widetilde{x}_{s-1})$, where $\widetilde{z}_{s-1}$ is the current more "aggressively" updated solution rather than $\widetilde{x}_{s-1}$; thus, this term also can be interpreted as momentum[5]. Then, we feed $\widetilde{y}_s$ to One Stage Accelerated SVRDA (Algorithm 7) as an initial point. Note that Algorithm 6 can be seen as a direct generalization of APG, because if we set $m = 1$, One Stage Accelerated SVRDA is essentially the same as one iteration PG with initial point $\widetilde{y}_s$; then, we can see that $\widetilde{z}_s = \widetilde{x}_s$, and Algorithm 6 essentially matches with deterministic APG. Next, we move to One Stage Accelerated SVRDA (Algorithm 7). Algorithm 7 is essentially a combination of the "accelerated regularized dual averaging" (AccSDA) method [21] with the variance reduction technique of SVRG. It updates $z_k$ by using the weighted average of all past variance reduced gradients $\bar{g}_k$ instead of only using a single variance reduced gradient $g_k$. Note that for constructing variance reduced gradient $g_k$, we use the full gradient of $F$ at $\widetilde{x}_{s-1}$ rather than the initial point $\widetilde{y}_s$. The

Adoption of (Innerly) Accelerated SVRDA rather than (Innerly) Accelerated SVRG enables us to construct lazy updates for sparse data (for more details, see Section E of supplementary material).

## 4.2 DASVRDA for strongly convex objectives

---

**Algorithm 8:** DASVRDA$^{\mathrm{sc}}(\check{x}_0, \gamma, \{L_i\}_{i=1}^n, m, b, S, T)$

---
    **for** $t = 1$ to $T$ **do**
       $\check{x}_t = \text{DASVRDA}^{\mathrm{ns}}(\check{x}_{t-1}, \check{x}_{t-1}, \gamma, \{L_i\}_{i=1}^n, m, b, S)$.
    **end for**
    **return** $\check{x}_T$.

---

Algorithm 8 is our proposed method for strongly convex objectives. Instead of directly accelerating the algorithm using a constant momentum rate, we restart Algorithm 6. Restarting scheme has several advantages both theoretically and practically. First, the restarting scheme only requires the optimal strong convexity of the objective instead of the ordinary strong convexity. Whereas, non-restarting accelerated algorithms essentially require the ordinary strong convexity of the objective. Second, for restarting algorithms, we can utilize adaptive restart schemes [14]. The adaptive restart schemes have been originally proposed for deterministic cases. The schemes are heuristic but quite effective empirically. The most fascinating property of these schemes is that we need not prespecify the strong convexity parameter $\mu$, and the algorithms adaptively determine the restart timings. [14] have proposed two heuristic adaptive restart schemes: the function scheme and gradient scheme. We can easily apply these ideas to our method, because our method is a direct generalization of the deterministic APG. For the function scheme, we restart Algorithm 6 if $P(\widetilde{x}_s) > P(\widetilde{x}_{s-1})$. For the gradient scheme, we restart the algorithm if $(\widetilde{y}_s - \widetilde{x}_s)^\top (\widetilde{y}_{s+1} - \widetilde{x}_s) > 0$. Here $\widetilde{y}_s - \widetilde{x}_s$ can be interpreted as a "one stage" gradient mapping of $P$ at $\widetilde{y}_s$. As $\widetilde{y}_{s+1} - \widetilde{x}_s$ is the momentum, this scheme can be interpreted such that we restart whenever the momentum and negative one Stage gradient mapping form an obtuse angle (this means that the momentum direction seems to be "bad"). We numerically demonstrate the effectiveness of these schemes in Section 6.

### Parameter tunings

For DASVRDA$^{\mathrm{ns}}$, only learning rate $\eta$ needs to be tuned, because we can theoretically obtain the optimal choice of $\gamma$, and we can naturally use $m = n/b$ as a default epoch length (see Section 5). For DASVRDA$^{\mathrm{sc}}$, both learning rate $\eta$ and fixed restart interval $S$ need to be tuned.

## 5 Convergence Analysis of DASVRDA Method

In this section, we provide the convergence analysis of our algorithms. Unless otherwise specified, serial computation is assumed. First, we consider the DASVRDA$^{\mathrm{ns}}$ algorithm.

**Theorem 5.1.** *Suppose that Assumptions 1, 2 and 3 hold. Let $\widetilde{x}_0, \widetilde{z}_0 \in \mathbb{R}^d$, $\gamma \geq 3$, $m \in \mathbb{N}$, $b \in [n]$ and $S \in \mathbb{N}$. Then DASVRDA$^{\mathrm{ns}}(\widetilde{x}_0, \widetilde{z}_0, \gamma, \{L_i\}_{i=1}^n, m, b, S)$ satisfies*

$$\mathbb{E}\left[P(\widetilde{x}_S) - P(x_*)\right] \leq \frac{4}{\left(1 - \frac{1}{\gamma}\right)^2 (S+2)^2} \left(\left(1 - \frac{1}{\gamma}\right)^2 (P(\widetilde{x}_0) - P(x_*)) + \frac{2}{\eta(m+1)m}\|\widetilde{z}_0 - x_*\|^2\right).$$

The proof of Theorem 5.1 is found in the supplementary material (Section A). We can easily see that the optimal choice of $\gamma$ is $(3 + \sqrt{9 + 8b/(m+1)})/2 = O(1 + b/m)$ (see Section B of supplementary material). We denote this value as $\gamma_*$. From Theorem 5.1, we obtain the following corollary:

**Corollary 5.2.** *Suppose that Assumptions 1, 2, and 3 hold. Let $\widetilde{x}_0 \in \mathbb{R}^d$, $\gamma = \gamma_*$, $m \propto n/b$ and $b \in [n]$. If we appropriately choose $S = O(\sqrt{(P(\widetilde{x}_0) - P(x_*))/\varepsilon} + (1/m + 1/\sqrt{mb})\sqrt{\bar{L}\|\widetilde{x}_0 - x_*\|^2/\varepsilon})$, then a total computational cost of DASVRDA$^{\mathrm{ns}}$ $(\widetilde{x}_0, \gamma_*, \{L_i\}_{i=1}^n, m, b, S)$ for $\mathbb{E}\left[P(\widetilde{x}_S) - P(x_*)\right] \leq \varepsilon$ is*

$$O\left(d\left(n\sqrt{\frac{P(\widetilde{x}_0) - P(x_*)}{\varepsilon}} + (b + \sqrt{n})\sqrt{\frac{\bar{L}\|\widetilde{x}_0 - x_*\|^2}{\varepsilon}}\right)\right).$$

*Remark.* If we adopt a warm start scheme for DASVRDA$^{\mathrm{ns}}$, we can further improve the rate to

$$O\left(d\left(n\log\left(\frac{P(\widetilde{x}_0) - P(x_*)}{\varepsilon}\right) + (b + \sqrt{n})\sqrt{\frac{L\|\widetilde{x}_0 - x_*\|^2}{\varepsilon}}\right)\right)$$

(see Section C and D of supplementary material).

Next, we analyze the DASVRDA$^{\mathrm{sc}}$ algorithm for optimally strongly convex objectives. Combining Theorem 5.1 with the optimal strong convexity of the objective function immediately yields the following theorem, which implies that the DASVRDA$^{\mathrm{sc}}$ algorithm achieves a linear convergence.

**Theorem 5.3.** *Suppose that Assumptions 1, 2, 3 and 4 hold. Let $\check{x}_0 \in \mathbb{R}^d$, $\gamma = \gamma_*$, $m \in \mathbb{N}$, $b \in [n]$ and $T \in \mathbb{N}$. Define $\rho \stackrel{\mathrm{def}}{=} 4\{(1 - 1/\gamma_*)^2 + 4/(\eta(m+1)m\mu)\}/\{(1 - 1/\gamma_*)^2(S+2)^2\}$. If $S$ is sufficiently large such that $\rho \in (0, 1)$, then DASVRDA$^{\mathrm{sc}}(\check{x}_0, \gamma_*, \{L_i\}_{i=1}^n, m, b, S, T)$ satisfies*

$$\mathbb{E}[P(\check{x}_T) - P(x_*)] \le \rho^T[P(\check{x}_0) - P(x_*)].$$

From Theorem 5.3, we have the following corollary.

**Corollary 5.4.** *Suppose that Assumptions 1, 2, 3 and 4 hold. Let $\check{x}_0 \in \mathbb{R}^d$, $\gamma = \gamma_*$, $m \propto n/b$, $b \in [n]$. There exists $S = O(1 + (b/n + 1/\sqrt{n})\sqrt{\overline{L}/\mu})$, such that $1/\log(1/\rho) = O(1)$. Moreover, if we appropriately choose $T = O(\log(P(\check{x}_0) - P(x_*)/\varepsilon)$, then a total computational cost of DASVRDA$^{\mathrm{sc}}(\check{x}_0, \gamma_*, \{L_i\}_{i=1}^n, m, b, S, T)$ for $\mathbb{E}[P(\check{x}_T) - P(x_*)] \le \varepsilon$ is*

$$O\left(d\left(n + (b + \sqrt{n})\sqrt{\frac{\overline{L}}{\mu}}\right)\log\left(\frac{P(\check{x}_0) - P(x_*)}{\varepsilon}\right)\right).$$

*Remark.* Corollary 5.4 implies that if the mini-batch size $b$ is $O(\sqrt{n})$, DASVRDA$^{\mathrm{sc}}(\check{x}_0, \gamma_*, \{L_i\}_{i=1}^n, n/b, b, S, T)$ still achieves the total computational cost of $O(d(n + \sqrt{n\overline{L}/\mu})\log(1/\varepsilon))$, which is much better than $O(d(n + \sqrt{nb\overline{L}/\mu})\log(1/\varepsilon))$ of APCG, SPDC, and Katyusha.

*Remark.* Corollary 5.4 also implies that DASVRDA$^{\mathrm{sc}}$ only needs size $O(\sqrt{n})$ mini-batches for achieving the optimal iteration complexity $O(\sqrt{L/\mu}\log(1/\varepsilon))$, when $L/\mu \ge n$. In contrast, APCG, SPDC and Katyusha need size $O(n)$ mini-batches and AccProxSVRG does $O(\sqrt{L/\mu})$ ones for achieving the optimal iteration complexity. Note that even when $L/\mu \le n$, our method only needs size $O(n\sqrt{\mu/L})$ mini-batches [6]. This size is smaller than $O(n)$ of APCG, SPDC, and Katyusha, and the same as that of AccProxSVRG.

# 6 Numerical Experiments

In this section, we provide numerical experiments to demonstrate the performance of DASVRDA.

We numerically compare our method with several well-known stochastic gradient methods in mini-batch settings: SVRG [22] (and SVRG$^{++}$ [2]), AccProxSVRG [12], Universal Catalyst [8], APCG [9], and Katyusha [1]. The details of the implemented algorithms and their parameter tunings are found in the supplementary material. In the experiments, we focus on the regularized logistic regression problem for binary classification, with regularizer $\lambda_1\|\cdot\|_1 + (\lambda_2/2)\|\cdot\|_2^2$.

We used three publicly available data sets in the experiments. Their sizes $n$ and dimensions $d$, and common min-batch sizes $b$ for all implemented algorithms are listed in Table 2.

Table 2: Summary of the data sets and mini-batch size used in our numerical experiments

| Data sets | $n$ | $d$ | $b$ |
|---|---|---|---|
| a9a | $32,561$ | $123$ | $180$ |
| rcv1 | $20,242$ | $47,236$ | $140$ |
| sido0 | $12,678$ | $4,932$ | $100$ |

For regularization parameters, we used three settings $(\lambda_1, \lambda_2) = (10^{-4}, 0)$, $(10^{-4}, 10^{-6})$, and $(0, 10^{-6})$. For the former case, the objective is non-strongly convex, and for the latter two cases,

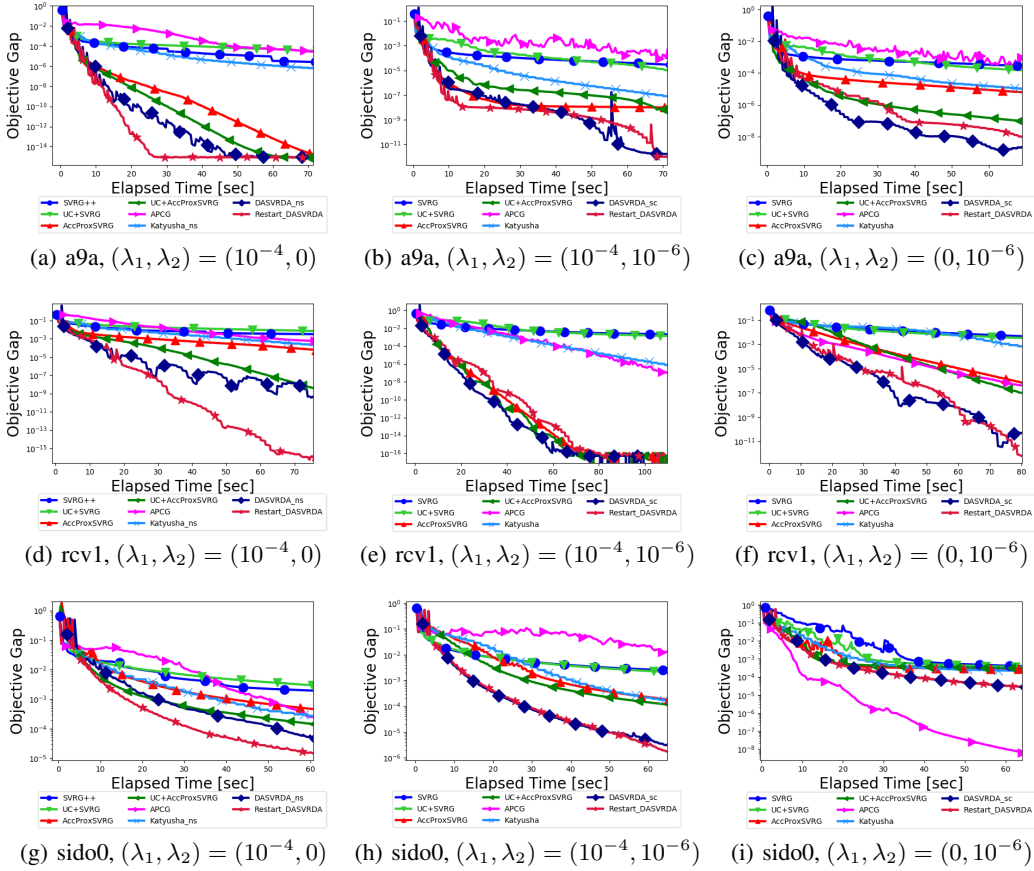

Figure 1: Comparisons on a9a (top), rcv1 (middle) and sido0 (bottom) data sets, for regularization parameters $(\lambda_1, \lambda_2) = (10^{-4}, 0)$ (left), $(\lambda_1, \lambda_2) = (10^{-4}, 10^{-6})$ (middle) and $(\lambda_1, \lambda_2) = (0, 10^{-6})$ (right).

the objectives are strongly convex. Note that for the latter two cases, the strong convexity of the objectives is $\mu = 10^{-6}$ and is relatively small; thus, it makes acceleration methods beneficial.

Figure 1 shows the comparisons of our method with the different methods described above on several settings. "Objective Gap" means $P(x) - P(x_*)$ for the output solution $x$. "Elapsed Time [sec]" means the elapsed CPU time (sec). "Restart_DASVRDA" means DASVRDA with heuristic adaptive restarting (Section 4). We can observe the following from these results:

- Our proposed DASVRDA and Restart DASVRDA significantly outperformed all the other methods overall.

- DASVRDA with the heuristic adaptive restart scheme efficiently made use of the local strong convexities of non-strongly convex objectives and significantly outperformed vanilla DASVRDA. For the other settings, the algorithm was still comparable to vanilla DASVRDA.

- UC+AccProxSVRG[7] outperformed vanilla AccProxSVRG but was outperformed by our methods overall.

- APCG sometimes performed unstably and was outperformed by vanilla SVRG. On sido0 data set, for Ridge Setting, APCG significantly outperformed all the other methods.
- Katyusha always outperformed vanilla SVRG, but was significantly outperformed by our methods.

## 7  Conclusion

In this paper, we developed a new accelerated stochastic variance reduced gradient method for regularized empirical risk minimization problems in mini-batch settings: DASVRDA. We have shown that DASVRDA achieves the total computational costs of $O(d(n\log(1/\varepsilon) + (b + \sqrt{n})\sqrt{L/\varepsilon}))$ and $O(d(n + (b + \sqrt{n})\sqrt{L/\mu})\log(1/\varepsilon))$ in size $b$ mini-batch settings for non-strongly and optimally strongly convex objectives, respectively. In addition, DASVRDA essentially achieves the optimal iteration complexities only with size $O(\sqrt{n})$ mini-batches for both settings. In the numerical experiments, our method significantly outperformed state-of-the-art methods, including Katyusha and AccProxSVRG.

## Acknowledgment

This work was partially supported by MEXT kakenhi (25730013, 25120012, 26280009 and 15H05707), JST-PRESTO and JST-CREST.

## Footnotes

[1] More precisely, the rate of ASDCA (UC) is with extra log-factors, and near but worse than the one of APCG, SPDC and Katyusha. This means that ASDCA (UC) cannot be optimal.

[2] Katyusha also achieves a near optimal total computational cost for non-strongly convex objectives.

[3] We refer to "optimal iteration complexity" as the iteration complexity of deterministic Nesterov's acceleration method [11].

[4] Actually, when $L/\varepsilon \leq n$ and $L/\mu \leq n$, our method needs size $O(n\sqrt{\varepsilon/L})$ and $O(n\sqrt{\mu/L})$ mini-batches, respectively, which are larger than $O(\sqrt{n})$, but smaller than $O(n)$. Achieving optimal iteration complexity for solving high accuracy and bad conditioned problems is much more important than doing ones with low accuracy and well-conditioned ones, because the former needs more overall computational cost than the latter.

[5]This form also arises in Monotone APG [7]. In Algorithm 7, $\widetilde{x} = x_m$ can be rewritten as $(2/(m(m+1)))\sum_{k=1}^m kz_k$, which is a weighted average of $z_k$; thus, we can say that $\widetilde{z}$ is updated more "aggressively" than $\widetilde{x}$. For the outerly accelerated SVRG (that is a combination of Algorithm 6 with vanilla SVRG, see section 3), $\widetilde{z}$ and $\widetilde{x}$ correspond to $x_m$ and $(1/m)\sum_{k=1}^m x_k$ in [22], respectively. Thus, we can also see that $\widetilde{z}$ is updated more "aggressively" than $\widetilde{x}$.

[6]Note that the required size is $O(n\sqrt{\mu/L})(\le O(n))$, which is not $O(n\sqrt{L/\mu}) \ge O(n)$.

[7]Although there has been no theoretical guarantee for UC + AccProxSVRG, we thought that it was fair to include experimental results about that because UC + AccProxSVRG gives better performances than the vanilla AccProxSVRG. Through some theoretical analysis, we can prove that UC + AccProxSVRG also has the similar rate and mini-batch efficiency to our proposed method, although these results are not obtained in any literature. However, our proposed method is superior to this algorithm both theoretically and practically, because the algorithm has several drawbacks due to the use of UC as follows. First, the algorithm has an additional logarithmic factor in its convergence rate. This factor is generally not negligible and slows down its practical performances. Second, the algorithm has more tuning parameters than our method. Third, the stopping criterion of each sub-problem of UC is hard to be tuned.

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
