[Supplementary Material · supple_dasvrda_nips2017_camera_ready.pdf]

# Supplementary material: Doubly Accelerated Stochastic Variance Reduced Dual Averaging Method for Regularized Empirical Risk Minimization

**Tomoya Murata**
NTT DATA Mathematical Systems Inc. , Tokyo, Japan
murata@msi.co.jp

**Taiji Suzuki**
Department of Mathematical Informatics
Graduate School of Information Science and Technology, The University of Tokyo, Tokyo, Japan
PRESTO, Japan Science and Technology Agency, Japan
Center for Advanced Integrated Intelligence Research, RIKEN, Tokyo, Japan
taiji@mist.i.u-tokyo.ac.jp

In this supplementary material, we provide the proof of Theorem 5.1 (Section A), the optimality of $\gamma_*$ (Section B), the algorithm of DASVRDA$^{\mathrm{ns}}$ with warm start (Section C) and its convergence analysis (Section D), the lazy update algorithm of AccSVRDA (Section E) and the experimental details (Section F). Finally, we briefly discuss DASVRG method, which is a variant of DASVRDA method (Section G).

## A   Proof of Theorem 5.1

In this section, we give the comprehensive proof of Theorem 5.1. First we analyze One Stage Accelerated SVRDA algorithm.

**Lemma A.1.** *The sequence $\{\theta_k\}_{k \geq 1}$ defined in Algorithm 7 satisfies*

$$\theta_k - 1 = \theta_{k-2}$$

*for $k \geq 1$, where $\theta_{-1} \overset{\text{def}}{=} 0$.*

*Proof.* Since $\theta_k = \frac{k+1}{2}$ for $k \geq 0$, we have that

$$\theta_k - 1 = \frac{k+1}{2} - 1 = \frac{k-1}{2} = \theta_{k-2}.$$

$\square$

**Lemma A.2.** *The sequence $\{\theta_k\}_{k \geq 1}$ defined in Algorithm 7 satisfies*

$$\theta_m \theta_{m-1} = \sum_{k=1}^{m} \theta_{k-1}.$$

$\square$

*Proof.* Observe that

$$\theta_m \theta_{m-1} = \frac{m(m+1)}{4} = \sum_{k=1}^{m} \frac{k}{2} = \sum_{k=1}^{m} \theta_{k-1}.$$

$\square$

**Lemma A.3.** *For every $x, y \in \mathbb{R}^d$,*

$$F(y) + \langle \nabla F(y), x - y \rangle + R(x) \leq P(x) - \frac{1}{2\bar{L}} \frac{1}{n} \sum_{i=1}^{n} \frac{1}{nq_i} \|\nabla f_i(x) - \nabla f_i(y)\|^2.$$

*Proof.* Since $f_i$ is convex and $L_i$-smooth, we have (see [7])

$$f_i(y) + \langle \nabla f_i(y), x - y \rangle \leq f_i(x) - \frac{1}{2L_i} \|\nabla f_i(x) - \nabla f_i(y)\|^2.$$

By the definition of $\{q_i\}$, summing this inequality from $i = 1$ to $n$ and dividing it by $n$ results in

$$F(y) + \langle \nabla F(y), x - y \rangle \leq F(x) - \frac{1}{2\bar{L}} \frac{1}{n} \sum_{i=1}^{n} \frac{1}{nq_i} \|\nabla f_i(x) - \nabla f_i(y)\|^2.$$

Adding $R(x)$ to the both sides of this inequality gives the desired result. $\qquad\square$

**Lemma A.4.**

$$\bar{g}_k = \frac{1}{\theta_k \theta_{k-1}} \sum_{k'=1}^{k} \theta_{k'-1} g_{k'} \ (k \geq 1).$$

*Proof.* For $k = 1$, $\bar{g}_1 = g_1 = \frac{1}{1 \cdot \frac{1}{2}} \sum_{k'=1}^{1} \frac{1}{2} \cdot g_{k'}$ by the definition of $\theta_0$.
Assume that the claim holds for some $k \geq 1$. Then

$$
\begin{aligned}
\bar{g}_{k+1} &= \left(1 - \frac{1}{\theta_{k+1}}\right) \bar{g}_k + \frac{1}{\theta_{k+1}} g_{k+1} \\
&= \left(1 - \frac{2}{k+2}\right) \frac{4}{(k+1)k} \sum_{k'=1}^{k} \theta_{k'-1} g_{k'} + \frac{2}{k+2} g_{k+1} \\
&= \frac{4}{(k+2)(k+1)} \sum_{k'=1}^{k+1} \theta_{k'-1} g_{k'} \\
&= \frac{1}{\theta_{k+1} \theta_k} \sum_{k'=1}^{k+1} \theta_{k'-1} g_{k'}.
\end{aligned}
$$

The first equality follows from the definition of $\bar{g}_{k+1}$. Second equality is due to the assumption of the induction. This finishes the proof for Lemma A.4. $\qquad\square$

Next we prove the following main lemma for One Stage Accelerated SVRDA. The proof is inspired by the one of AccSDA given in [9].

**Lemma A.5.** *Let $\eta < 1/\bar{L}$. For One Stage Accelerated SVRDA, we have that*

$$\mathbb{E}[P(x_m) - P(x)]$$

$$\leq \frac{2}{\eta(m+1)m} \|z_0 - x\|^2 - \frac{2}{\eta(m+1)m} \mathbb{E}\|z_m - x\|^2$$

$$+ \frac{2}{(m+1)m} \sum_{k=1}^{m} \left( \frac{(k+1)k\mathbb{E}\|g_k - \nabla F(y_k)\|^2}{4\left(\frac{1}{\eta} - \bar{L}\right)} - \frac{k}{2\bar{L}} \frac{1}{n} \sum_{i=1}^{n} \frac{1}{nq_i} \mathbb{E}\|\nabla f_i(y_k) - \nabla f_i(x)\|^2 \right),$$

*for any $x \in \mathbb{R}^d$, where the expectations are taken with respect to $I_k (1 \leq k \leq m)$.*

*Proof.* We define

$$\ell_k(x) = F(y_k) + \langle \nabla F(y_k), x - y_k \rangle + R(x),$$
$$\hat{\ell}_k(x) = F(y_k) + \langle g_k, x - y_k \rangle + R(x).$$

Observe that $\ell_k, \hat{\ell}_k$ is convex and $\ell_k \leq P$ by the convexity of $F$ and $R$. Moreover, for $k \geq 1$ we have that

$$\sum_{k'=1}^{k} \theta_{k'-1} \hat{\ell}_{k'}(z) = \sum_{k'=1}^{k} \theta_{k'-1} F(y_k) + \sum_{k'=1}^{k} \langle \theta_{k'-1} g_{k'}, z - y_{k'} \rangle + \sum_{k'=1}^{k} \theta_{k'-1} R(z)$$

$$= \langle \theta_k \theta_{k-1} \bar{g}_k, z \rangle + \theta_k \theta_{k-1} R(z) + \sum_{k'=1}^{k} \theta_{k'-1} F(y_k) - \sum_{k'=1}^{k} \theta_{k'-1} \langle g_{k'}, y_{k'} \rangle.$$

The second equality follows from Lemma A.4 and Lemma A.2. Thus we see that $z_k = \underset{z \in \mathbb{R}^d}{\operatorname{argmin}} \left\{ \sum_{k'=1}^{k} \theta_{k'-1} \hat{\ell}_{k'}(z) + \frac{1}{2\eta} \|z - z_0\|^2 \right\}$. Observe that $F$ is convex and $\bar{L}$-smooth. Thus we have that

$$F(x_k) \le F(y_k) + \langle \nabla F(y_k), x_k - y_k \rangle + \frac{\bar{L}}{2} \|x_k - y_k\|^2. \tag{1}$$

Hence we see that

$$
\begin{aligned}
P(x_k) &\le \ell_k(x_k) + \frac{\bar{L}}{2} \|x_k - y_k\|^2 \\
&= \ell_k \left( \left(1 - \frac{1}{\theta_k}\right) x_{k-1} + \frac{1}{\theta_k} z_k \right) + \frac{\bar{L}}{2} \left\| \left(1 - \frac{1}{\theta_k}\right) x_{k-1} + \frac{1}{\theta_k} z_k - y_k \right\|^2 \\
&\le \left(1 - \frac{1}{\theta_k}\right) \ell_k(x_{k-1}) + \frac{1}{\theta_k} \ell_k(z_k) + \frac{\bar{L}}{2\theta_k^2} \|z_k - z_{k-1}\|^2 \\
&\le \left(1 - \frac{1}{\theta_k}\right) P(x_{k-1}) + \frac{1}{\theta_k \theta_{k-1}} \left( \theta_{k-1} \hat{\ell}_k(z_k) + \frac{\bar{L}}{2} \|z_k - z_{k-1}\|^2 \right) \\
&\quad - \frac{1}{\theta_k} \langle g_k - \nabla F(y_k), z_k - y_k \rangle \\
&= \left(1 - \frac{1}{\theta_k}\right) P(x_{k-1}) + \frac{1}{\theta_k \theta_{k-1}} \left( \theta_{k-1} \hat{\ell}_k(z_k) + \frac{1}{2\eta} \|z_k - z_{k-1}\|^2 \right) \\
&\quad - \frac{1}{2\theta_k \theta_{k-1}} \left( \frac{1}{\eta} - \bar{L} \right) \|z_k - z_{k-1}\|^2 - \frac{1}{\theta_k} \langle g_k - \nabla F(y_k), z_k - z_{k-1} \rangle \\
&\quad - \frac{1}{\theta_k} \langle g_k - \nabla F(y_k), z_{k-1} - y_k \rangle.
\end{aligned}
$$

The first inequality follows from (1). The first equality is due to the definition of $x_k$. The second inequality is due to the convexity of $\ell_k$ and the definition of $y_k$. The third inequality holds because $\ell_k \le P$ and $\frac{1}{\theta_k^2} \le \frac{1}{\theta_k \theta_{k-1}}$.

Since $\frac{1}{\eta} > \bar{L}$, we have that

$$
\begin{aligned}
&- \frac{1}{2\theta_k \theta_{k-1}} \left( \frac{1}{\eta} - \bar{L} \right) \|z_k - z_{k-1}\|^2 - \frac{1}{\theta_k} \langle g_k - \nabla F(y_k), z_k - z_{k-1} \rangle \\
&\le \frac{1}{\theta_k} \frac{\theta_{k-1} \|g_k - \nabla F(y_k)\|^2}{2 \left( \frac{1}{\eta} - \bar{L} \right)} \\
&\le \frac{\|g_k - \nabla F(y_k)\|^2}{2 \left( \frac{1}{\eta} - \bar{L} \right)}.
\end{aligned}
$$

The first inequality is due to Young's inequality. The second inequality holds because $\theta_{k-1} \le \theta_k$.

Using this inequality, we get that

$$
\begin{aligned}
P(x_k) &\le \left(1 - \frac{1}{\theta_k}\right) P(x_{k-1}) + \frac{1}{\theta_k \theta_{k-1}} \left( \theta_{k-1} \hat{\ell}_k(z_k) + \frac{1}{2\eta} \|z_k - z_{k-1}\|^2 \right) \\
&\quad + \frac{\|g_k - \nabla F(y_k)\|^2}{2 \left( \frac{1}{\eta} - \bar{L} \right)} - \frac{1}{\theta_k} \langle g_k - \nabla F(y_k), z_{k-1} - y_k \rangle.
\end{aligned}
$$

Multiplying the both sides of the above inequality by $\theta_k \theta_{k-1}$ yields

$$
\begin{aligned}
\theta_k \theta_{k-1} P(x_k) &\le \theta_{k-1}(\theta_k - 1) P(x_{k-1}) + \theta_{k-1} \hat{\ell}_k(z_k) + \frac{1}{2\eta} \|z_k - z_{k-1}\|^2 \\
&\quad + \frac{\theta_k \theta_{k-1} \|g_k - \nabla F(y_k)\|^2}{2 \left( \frac{1}{\eta} - \bar{L} \right)} - \theta_{k-1} \langle g_k - \nabla F(y_k), z_{k-1} - y_k \rangle. \tag{2}
\end{aligned}
$$

By the fact that $\sum_{k'=1}^{k-1} \theta_{k'-1}\hat{\ell}_{k'}(z) + \frac{1}{2\eta}\|z - z_0\|^2$ is $\frac{1}{\eta}$-strongly convex and $z_{k-1}$ is the minimizer of $\sum_{k'=1}^{k-1} \theta_{k'-1}\hat{\ell}_{k'}(z) + \frac{1}{2\eta}\|z - z_0\|^2$ for $k \geq 2$, we have that

$$\sum_{k'=1}^{k-1} \theta_{k'-1}\hat{\ell}_{k'}(z_{k-1}) + \frac{1}{2\eta}\|z_{k-1} - z_0\|^2 + \frac{1}{2\eta}\|z_k - z_{k-1}\|^2 \leq \sum_{k'=1}^{k-1} \theta_{k'-1}\hat{\ell}_{k'}(z_k) + \frac{1}{2\eta}\|z_k - z_0\|^2$$

for $k \geq 1$ (and, for $k = 1$, we define $\sum_{k'=1}^{0} = 0$).

Using this inequality, we obtain

$$\theta_k\theta_{k-1}P(x_k) - \sum_{k'=1}^{k} \theta_{k'-1}\hat{\ell}_{k'}(z_k) - \frac{1}{2\eta}\|z_k - z_0\|^2$$

$$\leq \theta_{k-1}\theta_{k-2}P(x_{k-1}) - \sum_{k'=1}^{k-1} \theta_{k'-1}\hat{\ell}_{k'}(z_{k-1}) - \frac{1}{2\eta}\|z_{k-1} - z_0\|^2 + \frac{\theta_k\theta_{k-1}}{2\left(\frac{1}{\eta} - \bar{L}\right)}\|g_k - \nabla F(y_k)\|^2$$

$$- \theta_{k-1}\langle g_k - \nabla F(y_k), z_{k-1} - y_k\rangle.$$

Here, the inequality follows from Lemma A.1 (we defined $\theta_{-1} \stackrel{\text{def}}{=} 0$).

Summing the above inequality from $k = 1$ to $m$ results in

$$\theta_m\theta_{m-1}P(x_m) - \sum_{k=1}^{m} \theta_{k-1}\hat{\ell}_k(z_m) - \frac{1}{2\eta}\|z_m - z_0\|^2$$

$$\leq \sum_{k=1}^{m} \frac{\theta_k\theta_{k-1}\|g_k - \nabla F(y_k)\|^2}{2\left(\frac{1}{\eta} - \bar{L}\right)} - \sum_{k=1}^{m} \theta_{k-1}\langle g_k - \nabla F(y_k), z_{k-1} - y_k\rangle.$$

Using $\frac{1}{\eta}$-strongly convexity of the function $\sum_{k=1}^{m} \theta_{k-1}\hat{\ell}_k(z) + \frac{1}{2\eta}\|z - z_0\|^2$ and the optimality of $z_m$, we have that

$$\sum_{k=1}^{m} \theta_{k-1}\hat{\ell}_k(z_m) + \frac{1}{2\eta}\|z_m - z_0\|^2 \leq \sum_{k=1}^{m} \theta_{k-1}\hat{\ell}_k(x) + \frac{1}{2\eta}\|z_0 - x\|^2 - \frac{1}{2\eta}\|z_m - x\|^2.$$

From this inequality, we see that

$$\theta_m\theta_{m-1}P(x_m)$$

$$\leq \sum_{k=1}^{m} \theta_{k-1}\hat{\ell}_k(x) + \frac{1}{2\eta}\|z_0 - x\|^2 - \frac{1}{2\eta}\|z_m - x\|^2$$

$$+ \sum_{k=1}^{m} \frac{\theta_k\theta_{k-1}\|g_k - \nabla F(y_k)\|^2}{2\left(\frac{1}{\eta} - \bar{L}\right)} - \sum_{k=1}^{m} \theta_{k-1}\langle g_k - \nabla F(y_k), z_{k-1} - y_k\rangle$$

$$= \sum_{k=1}^{m} \theta_{k-1}\ell_k(x) + \frac{1}{2\eta}\|z_0 - x\|^2 - \frac{1}{2\eta}\|z_m - x\|^2$$

$$+ \sum_{k=1}^{m} \frac{\theta_k\theta_{k-1}\|g_k - \nabla F(y_k)\|^2}{2\left(\frac{1}{\eta} - \bar{L}\right)} - \sum_{k=1}^{m} \theta_{k-1}\langle g_k - \nabla F(y_k), z_{k-1} - x\rangle.$$

By Lemma A.3 with $x = x$ and $y = y_k$, we have that

$$\ell_k(x) \leq P(x) - \frac{1}{2\bar{L}}\frac{1}{n}\sum_{i=1}^{n} \frac{1}{nq_i}\|\nabla f_i(x) - \nabla f_i(y_k)\|^2.$$

Applying this inequality to the above inequality yields

$$\theta_m\theta_{m-1}P(x_m) - \sum_{k=1}^m \theta_{k-1}P(x)$$

$$\leq \frac{1}{2\eta}\|z_0 - x\|^2 - \frac{1}{2\eta}\|z_m - x\|^2$$

$$+ \sum_{k=1}^m \left[ \frac{\theta_k\theta_{k-1}\|g_k - \nabla F(y_k)\|^2}{2\left(\frac{1}{\eta} - \bar{L}\right)} - \frac{\theta_{k-1}}{2\bar{L}}\frac{1}{n}\sum_{i=1}^n \frac{1}{nq_i}\|\nabla f_i(x) - \nabla f_i(y_k)\|^2 \right]$$

$$- \sum_{k=1}^m \theta_{k-1}\langle g_k - \nabla F(y_k), z_{k-1} - x\rangle.$$

Using Lemma A.2 and dividing the both sides of the above inequality by $\theta_m\theta_{m-1}$ result in

$$P(x_m) - P(x)$$

$$\leq \frac{1}{2\eta\theta_m\theta_{m-1}}\|z_0 - x\|^2 - \frac{1}{2\eta\theta_m\theta_{m-1}}\|z_m - x\|^2$$

$$+ \frac{1}{\theta_m\theta_{m-1}}\sum_{k=1}^m \left[ \frac{\theta_k\theta_{k-1}\|g_k - \nabla F(y_k)\|^2}{2\left(\frac{1}{\eta} - \bar{L}\right)} - \frac{\theta_{k-1}}{2\bar{L}}\frac{1}{n}\sum_{i=1}^n \frac{1}{nq_i}\|\nabla f_i(x) - \nabla f_i(y_k)\|^2 \right]$$

$$- \frac{1}{\theta_m\theta_{m-1}}\sum_{k=1}^m \theta_{k-1}\langle g_k - \nabla F(y_k), z_{k-1} - x\rangle.$$

Taking the expectations with respect to $I_k (1 \leq k \leq m)$ on the both sides of this inequality yields

$$\mathbb{E}[P(x_m) - P(x)]$$

$$\leq \frac{1}{2\eta\theta_m\theta_{m-1}}\|z_0 - x\|^2 - \frac{1}{2\eta\theta_m\theta_{m-1}}\mathbb{E}\|z_m - x\|^2$$

$$+ \frac{1}{\theta_m\theta_{m-1}}\sum_{k=1}^m \left[ \frac{\theta_k\theta_{k-1}\mathbb{E}\|g_k - \nabla F(y_k)\|^2}{2\left(\frac{1}{\eta} - \bar{L}\right)} - \frac{\theta_{k-1}}{2\bar{L}}\frac{1}{n}\sum_{i=1}^n \frac{1}{nq_i}\mathbb{E}\|\nabla f_i(x) - \nabla f_i(y_k)\|^2 \right].$$

Here we used the fact that $\mathbb{E}[g_k - \nabla F(y_k)] = 0$ for $k = 1, \ldots, m$. This finishes the proof of Lemma A.5. $\qquad\square$

Now we need the following lemma.

**Lemma A.6.** *For every $x \in \mathbb{R}^d$,*

$$\frac{1}{n}\sum_{i=1}^n \frac{1}{nq_i}\|\nabla f_i(x) - \nabla f_i(x_*)\|^2 \leq 2\bar{L}(P(x) - P(x_*)).$$

*Proof.* From the argument of the proof of Lemma A.3, we have

$$\frac{1}{n}\sum_{i=1}^n \frac{1}{nq_i}\|\nabla f_i(x) - \nabla f_i(x_*)\|^2 \leq 2\bar{L}(F(x) - \langle \nabla F(x_*), x - x_*\rangle - F(x_*)).$$

By the optimality of $x_*$, there exists $\xi_* \in \partial R(x_*)$ such that $\nabla F(x_*) + \xi_* = 0$. Then we have

$$-\langle \nabla F(x_*), x - x_*\rangle = \langle \xi_*, x - x_*\rangle \leq R(x) - R(x_*),$$

and hence

$$\frac{1}{n}\sum_{i=1}^n \frac{1}{nq_i}\|\nabla f_i(x) - \nabla f_i(x_*)\|^2 \leq 2\bar{L}(P(x) - P(x_*)).$$

$\qquad\square$

**Proposition A.7.** *Let $\gamma > 1$ and $\eta \leq 1/((1 + \gamma(m+1)/b)\bar{L})$. For One Pass Accelerated SVRDA, it follows that*

$$\mathbb{E}[P(x_m) - P(\widetilde{x})] \leq \frac{2}{\eta(m+1)m}\|\widetilde{y} - \widetilde{x}\|^2 - \frac{2}{\eta(m+1)m}\mathbb{E}\|z_m - \widetilde{x}\|^2,$$

*and*

$$\mathbb{E}[P(x_m) - P(x_*)]$$
$$\leq \frac{1}{\gamma}(P(\widetilde{x}) - P(x_*)) + \frac{2}{\eta(m+1)m}\|\widetilde{y} - x_*\|^2 - \frac{2}{\eta(m+1)m}\mathbb{E}\|z_m - x_*\|^2,$$

*where the expectations are taken with respect to $I_k(1 \leq k \leq m)$.*

*Proof.* We bound the variance of the averaged stochastic gradient $\mathbb{E}\|g_k - \nabla F(y_k)\|^2$:

$$\mathbb{E}\|g_k - \nabla F(y_k)\|^2$$
$$= \mathbb{E}\left[\mathbb{E}_{I_k}\|g_k - \nabla F(y_k)\|^2 \mid [k-1]\right]$$
$$= \frac{1}{b}\mathbb{E}\left[\mathbb{E}_{i\sim Q}\|(\nabla f_i(y_k) - \nabla f_i(\widetilde{x}))/nq_i + \nabla F(\widetilde{x}) - \nabla F(y_k)\|^2 \mid [k-1]\right]$$
$$\leq \frac{1}{b}\mathbb{E}\left[\mathbb{E}_{i\sim Q}\|(\nabla f_i(y_k) - \nabla f_i(\widetilde{x}))/nq_i\|^2 \mid [k-1]\right]$$
$$= \frac{1}{b}\mathbb{E}\left[\frac{1}{n}\sum_{i=1}^{n}\frac{1}{nq_i}\|\nabla f_i(y_k) - \nabla f_i(\widetilde{x})\|^2\right] \tag{3}$$
$$\leq \frac{2}{b}\mathbb{E}\left[\frac{1}{n}\sum_{i=1}^{n}\frac{1}{nq_i}\|\nabla f_i(y_k) - \nabla f_i(x_*)\|^2\right]$$
$$+ \frac{2}{b}\mathbb{E}\left[\frac{1}{n}\sum_{i=1}^{n}\frac{1}{nq_i}\|\nabla f_i(\widetilde{x}) - \nabla f_i(x_*)\|^2\right]$$
$$\leq \frac{2}{b}\mathbb{E}\left[\frac{1}{n}\sum_{i=1}^{n}\frac{1}{nq_i}\|\nabla f_i(y_k) - \nabla f_i(x_*)\|^2\right] + \frac{4\bar{L}}{b}(P(\widetilde{x}) - P(x_*)). \tag{4}$$

The second equality follows from the independency of the random variables $\{i_1, \ldots, i_b\}$ and the unbiasedness of $(\nabla f_i(y_k) - \nabla f_i(\widetilde{x}))/nq_i + \nabla F(\widetilde{x})$. The first inequality is due to the fact that $\mathbb{E}\|X - \mathbb{E}[X]\|^2 \leq \mathbb{E}\|X\|^2$. The second inequality follows from Young's inequality. The final inequality is due to Lemma A.6.

Since $\frac{1}{\eta} \geq \left(1 + \frac{\gamma(m+1)}{b}\right)\bar{L}$ and $\gamma > 1$, using (3) yields

$$\frac{(k+1)k}{4\left(\frac{1}{\eta} - \bar{L}\right)}\mathbb{E}\|g_k - \nabla F(y_k)\|^2 - \frac{k}{2\bar{L}}\mathbb{E}\left[\frac{1}{n}\sum_{i=1}^{n}\frac{1}{nq_i}\|\nabla f_i(y_k) - \nabla f_i(\widetilde{x})\|^2\right] \leq 0.$$

By Lemma A.5 (with $x = \widetilde{x}$) we have

$$\mathbb{E}[P(x_m) - P(\widetilde{x})] \leq \frac{2}{\eta(m+1)m}\|\widetilde{y} - \widetilde{x}\|^2 - \frac{2}{\eta(m+1)m}\mathbb{E}\|z_m - \widetilde{x}\|^2.$$

Similarly, combining Lemma A.5 (with $x = x_*$) with (4) results in

$$\mathbb{E}[P(x_m) - P(x_*)]$$
$$\leq \frac{1}{\gamma}(P(\widetilde{x}) - P(x_*)) + \frac{2}{\eta(m+1)m}\|\widetilde{y} - x_*\|^2 - \frac{2}{\eta(m+1)m}\mathbb{E}\|z_m - x_*\|^2.$$

These are the desired results. $\qquad\square$

**Lemma A.8.** *The sequence $\{\widetilde{\theta}_s\}_{s\geq 1}$ defined in Algorithm 6 satisfies*

$$\widetilde{\theta}_s\left(\widetilde{\theta}_s - 1 + \frac{1}{\gamma}\right) \leq \widetilde{\theta}_{s-1}^2$$

*for any $s \geq 1$.*

*Proof.* Since $\widetilde{\theta}_s = \left(1 - \frac{1}{\gamma}\right)\frac{s+2}{2}$ for $s \geq 0$, we have

$$\widetilde{\theta}_s \left(\widetilde{\theta}_s - 1 + \frac{1}{\gamma}\right)$$

$$= \left(1 - \frac{1}{\gamma}\right)\frac{s+2}{2}\left(\left(1 - \frac{1}{\gamma}\right)\frac{s+2}{2} - 1 + \frac{1}{\gamma}\right)$$

$$= \left(1 - \frac{1}{\gamma}\right)^2 \frac{s(s+2)}{4}$$

$$\leq \widetilde{\theta}_{s-1}^2.$$

This finishes the proof of Lemma A.8. $\qquad\qquad\square$

Now we are ready to proof Theorem 5.1.

*Proof of Theorem 5.1.* By Proposition A.7, we have

$$\mathbb{E}[P(\widetilde{x}_s) - P(\widetilde{x}_{s-1})] \leq \frac{2}{\eta(m+1)m}\mathbb{E}\|\widetilde{y}_s - \widetilde{x}_{s-1}\|^2 - \frac{2}{\eta(m+1)m}\mathbb{E}\|\widetilde{z}_s - \widetilde{x}_{s-1}\|^2,$$

and

$$\mathbb{E}[P(\widetilde{x}_s) - P(x_*)]$$
$$\leq \frac{1}{\gamma}\mathbb{E}[P(\widetilde{x}_{s-1}) - P(x_*)] + \frac{2}{\eta(m+1)m}\mathbb{E}\|\widetilde{y}_s - x_*\|^2 - \frac{2}{\eta(m+1)m}\mathbb{E}\|\widetilde{z}_s - x_*\|^2,$$

where the expectations are taken with respect to the history of all random variables.

Hence we have

$$\mathbb{E}[P(\widetilde{x}_s) - P(\widetilde{x}_{s-1})] \leq \frac{4}{\eta(m+1)m}\mathbb{E}\langle\widetilde{z}_s - \widetilde{y}_s, \widetilde{x}_{s-1} - \widetilde{y}_s\rangle - \frac{2}{\eta(m+1)m}\mathbb{E}\|\widetilde{z}_s - \widetilde{y}_s\|^2, \quad (5)$$

and

$$\mathbb{E}[P(\widetilde{x}_s) - P(x_*)]$$
$$\leq \frac{1}{\gamma}\mathbb{E}[P(\widetilde{x}_{s-1}) - P(x_*)] + \frac{4}{\eta(m+1)m}\mathbb{E}\langle\widetilde{z}_s - \widetilde{y}_s, x_* - \widetilde{y}_s\rangle - \frac{2}{\eta(m+1)m}\mathbb{E}\|\widetilde{z}_s - \widetilde{y}_s\|^2. \quad (6)$$

Since $\gamma \geq 3$, we have $\widetilde{\theta}_s \geq 1$ for $s \geq 1$. Multiplying (5) by $\widetilde{\theta}_s(\widetilde{\theta}_s - 1) \geq 0$ and adding $\widetilde{\theta}_s \times$ (6) yield

$$\widetilde{\theta}_s^2\mathbb{E}[P(\widetilde{x}_s) - P(x_*)] - \widetilde{\theta}_s\left(\widetilde{\theta}_s - 1 + \frac{1}{\gamma}\right)\mathbb{E}[P(\widetilde{x}_{s-1}) - P(x_*)]$$

$$\leq \frac{4}{\eta(m+1)m}\mathbb{E}\langle\widetilde{\theta}_s(\widetilde{z}_s - \widetilde{y}_s), (\widetilde{\theta}_s - 1)\widetilde{x}_{s-1} - \widetilde{\theta}_s\widetilde{y}_s + x_*\rangle - \frac{2}{\eta(m+1)m}\mathbb{E}\|\widetilde{\theta}_s(\widetilde{z}_s - \widetilde{y}_s)\|^2.$$

By Lemma (A.8), we have

$$\widetilde{\theta}_s\left(\widetilde{\theta}_s - 1 + \frac{1}{\gamma}\right) \leq \widetilde{\theta}_{s-1}^2$$

for $s \geq 1$.

Thus we get

$$\widetilde{\theta}_s^2\mathbb{E}[P(\widetilde{x}_s) - P(x_*)] - \widetilde{\theta}_{s-1}^2\mathbb{E}[P(\widetilde{x}_{s-1}) - P(x_*)]$$

$$\leq \frac{4}{\eta(m+1)m}\mathbb{E}\langle\widetilde{\theta}_s(\widetilde{z}_s - \widetilde{y}_s), (\widetilde{\theta}_s - 1)\widetilde{x}_{s-1} - \widetilde{\theta}_s\widetilde{y}_s + x_*\rangle - \frac{2}{\eta(m+1)m}\mathbb{E}\|\widetilde{\theta}_s(\widetilde{z}_s - \widetilde{y}_s)\|^2.$$

$$= \frac{2}{\eta(m+1)m}\left(\mathbb{E}\|(\widetilde{\theta}_s - 1)\widetilde{x}_{s-1} - \widetilde{\theta}_s\widetilde{y}_s + x_*\|^2 - \mathbb{E}\|(\widetilde{\theta}_s - 1)\widetilde{x}_{s-1} - \widetilde{\theta}_s\widetilde{z}_s + x_*\|^2\right)$$

Since $\widetilde{y}_s = \widetilde{x}_{s-1} + \frac{\widetilde{\theta}_{s-1}-1}{\widetilde{\theta}_s}(\widetilde{x}_{s-1} - \widetilde{x}_{s-2}) + \frac{\widetilde{\theta}_{s-1}}{\widetilde{\theta}_s}(\widetilde{z}_{s-1} - \widetilde{x}_{s-1})$, we have

$$(\widetilde{\theta}_s - 1)\widetilde{x}_{s-1} - \widetilde{\theta}_s\widetilde{y}_s + x_* = (\widetilde{\theta}_{s-1} - 1)\widetilde{x}_{s-2} - \widetilde{\theta}_{s-1}\widetilde{z}_{s-1} + x_*.$$

Therefore summing the above inequality from $s = 1$ to $S$, we obtain

$$\widetilde{\theta}_s^2 \mathbb{E}[P(\widetilde{x}_S) - P(x_*)]$$

$$\leq \widetilde{\theta}_0^2 (P(\widetilde{x}_0) - P(x_*)) + \frac{2}{\eta(m+1)m} \|(\widetilde{\theta}_0 - 1)\widetilde{x}_{-1} - \widetilde{\theta}_0 \widetilde{z}_0 + x_*\|^2$$

$$= \left(1 - \frac{1}{\gamma}\right)^2 (P(\widetilde{x}_0) - P(x_*)) + \frac{2}{\eta(m+1)m} \|\widetilde{z}_0 - x_*\|^2.$$

Dividing both sides by $\widetilde{\theta}_s^2$ finishes the proof of Theorem 5.1.  □

## B    Optimal choice of $\gamma$

In this section, we prove the optimality of $\gamma_*$. We can choose the optimal value of $\gamma$ based on the following lemma.

**Lemma B.1.** *Define* $g(\gamma) \stackrel{\text{def}}{=} \frac{\left(1 + \frac{\gamma(m+1)}{b}\right)}{\left(1 - \frac{1}{\gamma}\right)^2}$ *for* $\gamma > 1$. *Then,*

$$\gamma_* \stackrel{\text{def}}{=} \operatorname*{argmin}_{\gamma > 1} g(\gamma) = \frac{1}{2}\left(3 + \sqrt{9 + \frac{8b}{m+1}}\right).$$

*Proof.* First observe that

$$g'(\gamma) = \frac{\frac{m+1}{b}\left(1 - \frac{1}{\gamma}\right)^2 - 2\left(1 + \frac{\gamma(m+1)}{b}\right)\left(1 - \frac{1}{\gamma}\right)\frac{1}{\gamma^2}}{\left(1 - \frac{1}{\gamma}\right)^2}.$$

Hence we have

$$g'(\gamma) = 0$$

$$\iff \frac{m+1}{b}\left(1 - \frac{1}{\gamma}\right)^2 - 2\left(1 + \frac{\gamma(m+1)}{b}\right)\left(1 - \frac{1}{\gamma}\right)\frac{1}{\gamma^2} = 0$$

$$\iff \frac{m+1}{b}(\gamma^2 - \gamma) - 2\left(1 + \frac{\gamma(m+1)}{b}\right) = 0$$

$$\iff \gamma^2 - 3\gamma - \frac{2b}{m+1} = 0$$

$$\iff \gamma = \frac{1}{2}\left(3 + \sqrt{9 + \frac{8b}{m+1}}\right) = \gamma_*.$$

Here the second and last equivalencies hold from $\gamma > 1$. Moreover observe that $g'(\gamma) > 0$ for $\gamma > \gamma_*$ and $g'(\gamma) < 0$ for $1 < \gamma < \gamma_*$. This means that $\gamma_*$ is the minimizer of $g$ on the region $\gamma > 1$.  □

## C    DASVRDA$^{\text{ns}}$ with warm start

In this section, we provide the algorithm of DASVRDA$^{\text{ns}}$ with warm start and its convergence analysis. First we describe the details of the algorithm. Algorithms 1 is a combination of DASVRDA$^{\text{ns}}$ with warm start scheme. At the warm start phase, we repeatedly run One Stage AccSVRDA and increment $m_u$ exponentially until $m_u \propto m$, where $m$ is the number of the inner iterations of DASVRDA$^{\text{ns}}$. After that, we run vanilla DASVRDA$^{\text{ns}}$. This algorithm gives a faster rate than vanilla DASVRDA$^{\text{ns}}$.
*Remark.* For DASVRDA$^{\text{sc}}$, the warm start scheme for DASVRDA$^{\text{ns}}$ is not needed because the theoretical rate is identical to the one without warm start.

**Theorem C.1.** *Suppose that Assumptions 1, 2 and 3 hold. Let* $\widetilde{x}_0 \in \mathbb{R}^d$, $\gamma = \gamma_*$, $m \in \mathbb{N}$, $m_0 = \min\left\{\left\lceil \sqrt{(1 + \gamma(m+1)/b)\bar{L}\frac{\|\widetilde{x}_0 - x_*\|^2}{P(\widetilde{x}_0) - P(x_*)}} \right\rceil, m\right\} \in \mathbb{N}$, $b \in [n]$, $U = \lceil \log_{\sqrt{\gamma}}(m/m_0) \rceil$ *and* $S \in \mathbb{N}$. *Then DASVRDA$^{\text{ns}}$ with warm start$(\widetilde{x}_0, \gamma_*, \{L_i\}_{i=1}^n, m_0, m, b, U, S)$ satisfies*

$$\mathbb{E}\left[P(\widetilde{x}_S) - P(x_*)\right] \leq O\left(\frac{1}{S^2}\left(\frac{1}{m^2} + \frac{1}{mb}\right)\bar{L}\|\widetilde{x}_0 - x_*\|^2\right).$$

---

**Algorithm 1:** DASVRDA$^{\mathrm{ns}}$ with warm start $(\widetilde{x}_0, \gamma, \{L_i\}_{i=1}^n, m_0, m, b, U, S)$

---

$\widetilde{z}_0 = \widetilde{x}_0,\ \bar{L} = \frac{1}{n}\sum_{i=1}^n L_i,\ Q = \{q_i\} = \left\{\frac{L_i}{n\bar{L}}\right\}$.

**for** $u = 1$ to $U$ **do**
$\quad m_u = \left\lceil \sqrt{\gamma(m_{u-1} + 1)m_{u-1}} \right\rceil$
**end for**
$m'_U = \left\lceil \sqrt{(m_U + 1)m_U/(1 - 1/\gamma)} \right\rceil$.
$\eta = \dfrac{1}{\left(1 + \frac{\gamma(m'_U + 1)}{b}\right)\bar{L}}$

**for** $u = 1$ to $U$ **do**
$\quad (\widetilde{x}_u, \widetilde{z}_u) = $ One Stage AccSVRDA$(\widetilde{z}_{u-1}, \widetilde{x}_{u-1}, \eta, m_u, b, Q)$.
**end for**
**return** DASVRDA$^{\mathrm{ns}}(\widetilde{x}_U, \widetilde{z}_U, \gamma, \{L_i\}_{i=1}^n, m'_U, b, S)$.

---

For the proof of Theorem C.1, see Section D). From Theorem C.1, we obtain the following corollary:

**Corollary C.2.** *Suppose that Assumptions 1, 2, and 3 hold. Let $\widetilde{x}_0 \in \mathbb{R}^d$, $\gamma = \gamma_*$, $m \propto n/b$, $m_0 = \min\left\{\left\lceil\sqrt{(1 + \gamma(m+1)/b)\bar{L}\frac{\|\widetilde{x}_0 - x_*\|^2}{P(\widetilde{x}_0) - P(x_*)}}\right\rceil, m\right\} \in \mathbb{N}$, $b \in [n]$ and $U = \lceil\log_{\sqrt{\gamma}}(m/m_0)\rceil$. If we appropriately choose $S = O(1 + (1/m + 1/\sqrt{mb})\sqrt{\bar{L}\|\widetilde{x}_0 - x_*\|^2/\varepsilon})$, then a total computational cost of DASVRDA$^{\mathrm{ns}}$ with warm start$(\widetilde{x}_0, \gamma_*, \{L_i\}_{i=1}^n, m_0, m, b, U, S)$ for $\mathbb{E}\left[P(\widetilde{x}_S) - P(x_*)\right] \le \varepsilon$ is*

$$O\left(d\left(n\log\left(\frac{P(\widetilde{x}_0) - P(x_*)}{\varepsilon}\right) + (b + \sqrt{n})\sqrt{\frac{\bar{L}\|\widetilde{x}_0 - x_*\|^2}{\varepsilon}}\right)\right).$$

For the proof of Corollary C.2, see Section D.

*Remark.* Corollary 5.2 implies that if the mini-batch size $b$ is $O(\sqrt{n})$, DASVRDA$^{\mathrm{ns}}$ with warm start$(\widetilde{x}_0, \gamma_*, \{L_i\}_{i=1}^n, m_0, n/b, b, U, S)$ still achieves the total computational cost of $O(d(n\log(1/\varepsilon) + \sqrt{n\bar{L}/\varepsilon}))$, which is better than $O(d(n\log(1/\varepsilon) + \sqrt{nb\bar{L}/\varepsilon}))$ of Katyusha.

*Remark.* Corollary 5.2 also implies that DASVRDA$^{\mathrm{ns}}$ with warm start only needs size $O(\sqrt{n})$ mini-batches for achieving the optimal iteration complexity of $O(\sqrt{L/\varepsilon})$, when $L/\varepsilon \ge n\log^2(1/\varepsilon)$. In contrast, Katyusha needs size $O(n)$ mini-batches for achieving the optimal iteration complexity. Note that even when $L/\varepsilon \le n\log^2(1/\varepsilon)$, our method only needs size $\widetilde{O}(n\sqrt{\varepsilon/L})$ mini-batches[1], that is typically smaller than $O(n)$ of Katyusha.

## D  Proof of Theorem C.1

In this section, we give proofs of Theorem C.1 and Corollary C.2.

*Proof of Theorem C.1.* Since $\eta = 1/((1 + \gamma(m'_U + 1)/b)\bar{L}) \le 1/((1 + \gamma(m_u + 1)/b)\bar{L})$, from Proposition A.7, we have

$$\mathbb{E}[P(\widetilde{x}_u) - P(x_*)] + \frac{2}{\eta(m_u + 1)m_u}\mathbb{E}\|\widetilde{z}_u - x_*\|^2$$
$$\le \frac{1}{\gamma}(P(\widetilde{x}_{u-1}) - P(x_*)) + \frac{2}{\eta(m_u + 1)m_u}\|\widetilde{z}_{u-1} - x_*\|^2$$
$$= \frac{1}{\gamma}\left(P(\widetilde{x}_{u-1}) - P(x_*) + \frac{2\gamma}{\eta(m_u + 1)m_u}\|\widetilde{z}_{u-1} - x_*\|^2\right).$$

Since $m_u = \lceil\sqrt{\gamma(m_{u-1} + 1)m_{u-1}}\rceil$, we have

$$\frac{2\gamma}{\eta(m_u + 1)m_u} \le \frac{2}{\eta(m_{u-1} + 1)m_{u-1}}.$$

Using this inequality, we obtain that

$$\mathbb{E}[P(\widetilde{x}_U) - P(x_*)] + \frac{2}{\eta(m_U + 1)m_U}\mathbb{E}\|\widetilde{z}_U - x_*\|^2$$

$$\leq \frac{1}{\gamma}\left(P(\widetilde{x}_{U-1}) - P(x_*) + \frac{2}{\eta(m_{U-1} + 1)m_{U-1}}\mathbb{E}\|\widetilde{z}_{U-1} - x_*\|^2\right)$$

$$\leq \cdots$$

$$\leq \frac{1}{\gamma^U}\left(P(\widetilde{x}_0) - P(x_*) + \frac{2}{\eta(m_0 + 1)m_0}\|\widetilde{z}_0 - x_*\|^2\right)$$

$$\leq \frac{1}{\gamma^U}\left(P(\widetilde{x}_0) - P(x_*) + \frac{2}{\eta(m_0 + 1)m_0}\|\widetilde{x}_0 - x_*\|^2\right)$$

$$= O\left(\frac{1}{\gamma^U}(P(\widetilde{x}_0) - P(x_*))\right).$$

The last equality is due to the definitions of $m_0$ and $\eta$, and the fact $m'_U = O(m_U) = O(\sqrt{\gamma}^U m_0) = O(m)$ (see the arguments in the proof of Corollary 5.2. Since

$$\left(1 - \frac{1}{\gamma}\right)^2 (m'_U + 1)m'_U \geq (m_U + 1)m_U,$$

we get

$$\mathbb{E}[P(\widetilde{x}_U) - P(x_*)] + \frac{2}{\left(1 - \frac{1}{\gamma}\right)^2 \eta(m'_U + 1)m'_U}\mathbb{E}\|\widetilde{z}_U - x_*\|^2$$

$$\leq O\left(\frac{1}{\gamma^U}(P(\widetilde{x}_0) - P(x_*))\right).$$

Using the definitions of $U$ and $m_0$ and combining this inequality with Theorem 5.1, we obtain that desired result. $\qquad\square$

*Proof of Corollary C.2.* Observe that the total computational cost at the warm start phase becomes

$$O\left(dnU + db\sum_{u=1}^{U} m_u\right).$$

Since $m_u \leq \sqrt{\gamma}m_{u-1} + \sqrt{\gamma} + 1 \leq \sqrt{\gamma}m_{u-1} + 2\sqrt{\gamma} \leq \sqrt{\gamma}^2 m_{u-2} + 2\sqrt{\gamma} + 2\sqrt{\gamma}^2 \leq \cdots \leq \sqrt{\gamma}^u m_0 + 2\sum_{u'=1}^{u}\sqrt{\gamma}^{u'} = O(\sqrt{\gamma}^u m_0)$, we have

$$O\left(dnU + db\sum_{u=1}^{U} m_u\right) = O\left(dnU + db\sqrt{\gamma}^U m_0\right).$$

Suppose that $m \geq m_0\sqrt{(P(\widetilde{x}_0) - P(x_*))/\varepsilon}$. Then, this condition implies $U = \lceil\log_{\sqrt{\gamma}}(m/m_0)\rceil \geq \log_\gamma((P(\widetilde{x}_0) - P(x_*))/\varepsilon)$. Hence we only need to run $u = O(\log_\gamma((P(\widetilde{x}_0) - P(x_*))/\varepsilon)) \leq U$ iterations at the warm start phase and running DASVRDA$^{\text{ns}}$ is not needed. Then the total computational cost becomes

$$O\left(d\left(n\log\frac{P(\widetilde{x}_0) - P(x_*)}{\varepsilon} + bm_0\sqrt{\frac{P(\widetilde{x}_0) - P(x_*)}{\varepsilon}}\right)\right) \leq O\left(d\left(n\log\frac{P(\widetilde{x}_0) - P(x_*)}{\varepsilon}\right)\right),$$

here we used $mb = O(n)$. Next, suppose that $m \leq m_0\sqrt{(P(\widetilde{x}_0) - P(x_*))/\varepsilon}$. In this case, the total computational cost at the warm start phase with full U iterations becomes

$$O\left(d\left(n\log\frac{m}{m_0} + mb\right)\right) \leq O\left(d\left(n\log\frac{P(\widetilde{x}_0) - P(x_*)}{\varepsilon}\right)\right).$$

Finally, using Theorem C.1 yields the desired total computational cost. $\qquad\square$

# E  Lazy Update Algorithm of DASVRDA Method

In this section, we discuss how to efficiently compute the updates of the DASVRDA algorithm for sparse data. Specifically, we derive lazy update rules of One Stage Accelerated SVRDA for the following empirical risk minimization problem:

$$\frac{1}{n}\sum_{i=1}^{n}\psi_i(a_i^\top x) + \lambda_1 \|x\|_1 + \frac{\lambda_2}{2}\|x\|_2^2, \quad \lambda_1, \lambda_2 \geq 0$$

For the sake of simplicity, we define the one dimensional soft-thresholding operator as follows:

$$\mathrm{soft}(z,\lambda) \overset{\mathrm{def}}{=} \mathrm{sign}\,(z)\max\{|z| - \lambda, 0\},$$

for $z \in \mathbb{R}$. Moreover, in this section, we denote $[z_1, z_2]$ as $\{z \in \mathbb{Z} \mid z_1 \leq z \leq z_2\}$ for integers $z_1, z_2 \in \mathbb{Z}$.

Originally, lazy update was proposed in online settings [3]. Generally, it is difficult for accelerated stochastic variance reduction methods to construct lazy update rules because (i) generally, variance reduced gradients are not sparse even if stochastic gradients are sparse; (ii) if we adopt the momentum scheme, the updated solution becomes a convex combination of previous solutions; and (iii) for non-strongly convex objectives, the momentum rate must not be constant. [4] have tackled the problem of (i) on non-accelerated settings and derived lazy update rules of the "mini-batch semi-stochastic gradient descent" (mS2GD) method. [1] has only mentioned that lazy updates can be applied to Katyusha but did not give explicit lazy update rules of Katyusha. Particularly, for non-strongly convex objectives, it seems to be difficult to derive lazy update rules owing to the difficulty of (iii). The reason we adopt the stochastic dual averaging scheme [9] rather than stochastic gradient descent for our method is to be able to overcome the difficulties faced in (i), (ii), and (iii). The lazy update rules of our method support both non-strongly and strongly convex objectives.

The explicit algorithm of the lazy updates for One Stage Accelerated SVRDA is given by Algorithm 2. Let us analyze the iteration cost of the algorithm. Suppose that each feature vector $a_i$ is sparse and the expected number of the nonzero elements is $O(d')$. First note that $|\mathcal{A}_k| = O(bd')$ expectedly if $d' \ll d$. For updating $x_{k-1}$, by Proposition E.1, we need to compute $\sum_{k' \in K_j^\pm} \theta_{k'-2}/(1 + \eta\theta_{k'-1}\theta_{k'-2}\lambda_2)$ and $\sum_{k' \in K_j^\pm} \theta_{k'-1}\theta_{k'-2}^2/(1 + \eta\theta_{k'-1}\theta_{k'-2}\lambda_2)$ for each $j \in \mathcal{A}_k$. For this, we first make lists $\{S_k\}_{k=1}^m = \{\sum_{k'=1}^k \theta_{k'-2}/(1 + \eta\theta_{k'-1}\theta_{k'-2}\lambda_2)\}_{k=1}^m$ and $\{S_k'\}_{k=1}^m = \{\sum_{k'=1}^k \theta_{k'-1}\theta_{k'-2}^2/(1 + \eta\theta_{k'-1}\theta_{k'-2}\lambda_2)\}_{k=1}^m$ before running the algorithm. This needs only $O(m)$. Note that these lists are not depend on coordinate $j$. Since $K_j^\pm$ are sets of continuous integers in $[k_j + 2, k]$ or unions of two sets of continuous integers in $[k_j + 2, k]$, we can efficiently compute the above sums. For example, if $K_j^+ = [k_j + 2, s_-] \cup [s_+, k]$ for some integers $s_\pm \in [k_j + 2, k]$, we can compute $\sum_{k' \in K_j^+} \theta_{k'-2}/(1 + \eta\theta_{k'-1}\theta_{k'-2}\lambda_2)$ as $S_{s_-} - S_{k_j+1} + S_k - S_{s_+-1}$ and this costs only $O(1)$. Thus, for computing $x_{k-1}$ and $y_k$, we need only $O(bd')$ computational cost. For computing $g_k$, we need to compute the inner product $a_i^\top y_k$ for each $i \in I_k$ and this costs $O(bd')$ expectedly. The expected cost of the rest of the updates is apparently $O(bd')$. Hence, the total expected iteration cost of our algorithm in serial settings becomes $O(bd')$ rather than $O(bd)$. Furthermore, we can extend our algorithm to parallel computing settings. Indeed, if we have $b$ processors, processor $b'$ runs on the set $\mathcal{A}_k^{b'} \overset{\mathrm{def}}{=} \{j \in [d] \mid a_{i_{b'},j} \neq 0\}$. Then the total iteration cost per processor becomes ideally $O(d')$. Generally the overlap among the sets $\mathcal{A}_k^{b'}$ may cause latency, however for sufficiently sparse data, this latency is negligible. The following proposition guarantees that Algorithm 2 is equivalent to Algorithm 7 when $R(x) = \lambda_1\|x\|_1 + (\lambda_2/2)\|x\|_2^2$.

**Proposition E.1.** *Suppose that* $R(x) = \lambda_1\|x\|_1 + \frac{\lambda_2}{2}\|x\|_2^2$ *with* $\lambda_1, \lambda_2 \geq 0$. *Let* $j \in [d]$, $k_j \in [m] \cup \{0\}$ *and* $k \geq k_j + 1$. *Assume that* $\nabla_j f_i(y_{k'}) = \nabla_j f_i(\widetilde{x}) = 0$ *for any* $i \in [b]$ *and* $k' \in [k_j + 1, k - 1]$.

**Algorithm 2:** Lazy Updates for One Stage AccSVRDA $(\widetilde{y}, \widetilde{x}, \eta, m, b, Q)$

$x_0 = z_0 = \widetilde{y}$.
$g_{0,j}^{\text{sum}} = 0 \ (j \in [d])$.
$\theta_0 = \frac{1}{2}$.
$k_j = 0 \ (j \in [d])$.
$\widetilde{\nabla} = \nabla F(\widetilde{x})$.
**for** $k = 1$ to $m$ **do**
    Sample independently $i_1, \ldots, i_b \in [1, n]$ according to $Q$, set $I_k = \{i_1, \ldots, i_b\}$.
    $\mathcal{A}_k = \{j \in [d] \mid \exists b' \in [b] : a_{i_{b'}, j} \neq 0\}$.
    $\theta_k = \frac{k+1}{2}$.
    **for** $j \in \mathcal{A}_k$ **do**
        Update $x_{k-1,j}, y_{k,j}$ as in Proposition E.1.
    **end for**
    **for** $j \in \mathcal{A}_k$ **do**
        $g_{k,j} = \frac{1}{b} \sum_{i \in I_t} \frac{1}{nq_i} \left( \psi_i'(a_i^\top y_k) a_{i,j} - \psi_i'(a_i^\top \widetilde{x}) a_{i,j} \right) + \widetilde{\nabla}_j$.
        $g_{k,j}^{\text{sum}} = g_{k_j,j}^{\text{sum}} + \theta_{k-1} g_{k,j} + \left( \theta_k \theta_{k-1} - \theta_{k_j} \theta_{k_j-1} \right) \widetilde{\nabla}_j$.
        $z_{k,j} = \frac{1}{1+\eta\theta_k\theta_{k-1}\lambda_2} \text{soft}(z_{0,j} - \eta g_{k,j}^{\text{sum}}, \eta\theta_k\theta_{k-1}\lambda_1)$.
        $x_{k,j} = \left(1 - \frac{1}{\theta_k}\right) x_{k-1,j} + \frac{1}{\theta_k} z_{k,j}$.
        $k_j = k$.
    **end for**
**end for**
**return** $(x_m, z_m)$.

*In Algorithm 7, the following results hold:*

$$x_{k-1,j} = \begin{cases} x_{0,j} & (k=1) \\ \frac{\theta_{k_j}\theta_{k_j-1}}{\theta_{k-1}\theta_{k-2}} x_{k_j,j} + \frac{1}{\theta_{k-1}\theta_{k-2}} \sum_{k' \in K_j^+} \frac{\theta_{k'-2}}{1+\eta\theta_{k'-1}\theta_{k'-2}\lambda_2}(z_{0,j} - M_{k',j}^+) & (k \geq 2) \\ \quad + \frac{1}{\theta_{k-1}\theta_{k-2}} \sum_{k' \in K_j^-} \frac{\theta_{k'-2}}{1+\eta\theta_{k'-1}\theta_{k'-2}\lambda_2}(z_{0,j} - M_{k',j}^-) \end{cases} ,$$

$$y_{k,j} = \begin{cases} x_{0,j} & (k=1) \\ \left(1 - \frac{1}{\theta_k}\right) x_{k-1,j} + \frac{1}{\theta_k} \frac{1}{1+\eta\theta_{k-1}\theta_{k-2}\lambda_2} \times & (k \geq 2) \\ \quad \text{soft}\left(z_{0,j} - \eta g_{k_j,j}^{\text{sum}} - \eta(\theta_{k-1}\theta_{k-2} - \theta_{k_j}\theta_{k_j-1})\widetilde{\nabla}_j, \eta\theta_{k-1}\theta_{k-2}\lambda_1\right) \end{cases} ,$$

*and*

$$z_{k,j} = \frac{1}{1+\eta\theta_k\theta_{k-1}\lambda_2} \text{soft}(z_{0,j} - \eta g_{k,j}^{\text{sum}}, \eta\theta_k\theta_{k-1}\lambda_1),$$

*where*

$$M_{k',j}^{\pm} \stackrel{\text{def}}{=} \eta\theta_{k'-1}\theta_{k'-2}(\widetilde{\nabla}_j \pm \lambda_1) + \eta g_{k_j,j}^{\text{sum}} - \eta\theta_{k_j}\theta_{k_j-1}\widetilde{\nabla}_j,$$

*and $K_j^{\pm} \subset [k_j+2, k]$ are defined as follows:*

*Let $c_1 \stackrel{\text{def}}{=} \frac{\eta\widetilde{\nabla}_j}{4}$, $c_2 \stackrel{\text{def}}{=} \frac{\eta\lambda_1}{4}$ and $c_3 \stackrel{\text{def}}{=} \eta g_{k_j,j}^{\text{sum}} - \eta\theta_{k_j}\theta_{k_j-1}\widetilde{\nabla}_j$ to simplify the notation. Note that $c_2 \geq 0$. Moreover, we define*

$$D_{\pm} \stackrel{\text{def}}{=} (c_1 \pm c_2)^2 + 4(c_1 \pm c_2)(z_{0,j} - c_3),$$

$$s_{\pm}^+ \stackrel{\text{def}}{=} \frac{c_1 + c_2 \pm \sqrt{D_+}}{c_1 + c_2},$$

$$s_{\pm}^- \stackrel{\text{def}}{=} \frac{c_1 - c_2 \pm \sqrt{D_-}}{c_1 - c_2},$$

*where if $s_{\pm}^{\pm}$ are not well defined, we simply assign $0$ (or any number) to $s_{\pm}^{\pm}$.*

1) *If $c_1 > c_2$, then*

$$K_j^+ \stackrel{\text{def}}{=} \begin{cases} \emptyset & (D_+ \leq 0) \\ [k_j+2,k] \cap [\lceil s_-^+ \rceil, \lfloor s_+^+ \rfloor] & (D_+ > 0) \end{cases},$$

$$K_j^- \stackrel{\text{def}}{=} \begin{cases} [k_j+2,k] & (D_- \leq 0) \\ [k_j+2, \lfloor s_-^- \rfloor] \cup [\lceil s_+^- \rceil, k] & (D_- > 0) \end{cases}.$$

2) *If $c_1 = c_2$, then*

$$K_j^+ \stackrel{\text{def}}{=} \begin{cases} \emptyset & (c_2 = 0 \wedge z_{0,j} \leq c_3) \\ [k_j+2,k] & (c_2 = 0 \wedge z_{0,j} > c_3) \\ \emptyset & (c_2 > 0 \wedge D_+ \leq 0) \\ [k_j+2,k] \cap [\lceil s_-^+ \rceil, \lfloor s_+^+ \rfloor] & (c_2 > 0 \wedge D_+ > 0) \end{cases},$$

$$K_j^- \stackrel{\text{def}}{=} \begin{cases} [k_j+2,k] & (z_{0,j} < c_3) \\ \emptyset & (z_{0,j} \geq c_3) \end{cases}.$$

3) *If $|c_1| < c_2$, then*

$$K_j^+ \stackrel{\text{def}}{=} \begin{cases} \emptyset & (D_+ \leq 0) \\ [k_j+2,k] \cap [\lceil s_-^+ \rceil, \lfloor s_+^+ \rfloor] & (D_+ > 0) \end{cases},$$

$$K_j^- \stackrel{\text{def}}{=} \begin{cases} \emptyset & (D_- \leq 0) \\ [k_j+2,k] \cap [\lceil s_-^- \rceil, \lfloor s_+^- \rfloor] & (D_- > 0) \end{cases}.$$

4) *If $c_1 = -c_2$, then*

$$K_j^+ \stackrel{\text{def}}{=} \begin{cases} \emptyset & (z_{0,j} \leq c_3) \\ [k_j+2,k] & (z_{0,j} > c_3) \end{cases}$$

$$K_j^- \stackrel{\text{def}}{=} \begin{cases} [k_j+2,k] & (c_2 = 0 \wedge z_{0,j} < c_3) \\ \emptyset & (c_2 = 0 \wedge z_{0,j} \geq c_3) \\ \emptyset & (c_2 > 0 \wedge D_- \leq 0) \\ [k_j+2,k] \cap [\lceil s_-^- \rceil, \lfloor s_+^- \rfloor] & (c_2 > 0 \wedge D_- > 0) \end{cases}.$$

5) *If $c_1 < -c_2$, then*

$$K_j^+ \stackrel{\text{def}}{=} \begin{cases} [k_j+2,k] & (D_+ \leq 0) \\ [k_j+2, \lfloor s_-^+ \rfloor] \cup [\lceil s_+^+ \rceil, k] & (D_+ > 0) \end{cases},$$

$$K_j^- \stackrel{\text{def}}{=} \begin{cases} \emptyset & (D_- \leq 0) \\ [k_j+2,k] \cap [\lceil s_-^- \rceil, \lfloor s_+^- \rfloor] & (D_- > 0) \end{cases}.$$

*Proof.* First we consider the case $k = 1$. Observe that

$$y_{1,j} = \left(1 - \frac{1}{\theta_1}\right) x_{0,j} + \frac{1}{\theta_1} z_{0,j} = z_{0,j} = x_{0,j},$$

and

$$\begin{aligned} z_{1,j} &= \frac{1}{1 + \eta \theta_1 \theta_0 \lambda_2} \mathrm{soft}\left(z_{0,j} - \eta \theta_1 \theta_0 \frac{1}{\theta_1} g_{1,j}, \eta \theta_1 \theta_0 \lambda_1\right) \\ &= \frac{1}{1 + \eta \theta_1 \theta_0 \lambda_2} \mathrm{soft}\left(z_{0,j} - \eta \theta_0 g_{1,j}, \eta \theta_1 \theta_0 \lambda_1\right) \\ &= \frac{1}{1 + \eta \theta_1 \theta_0 \lambda_2} \mathrm{soft}\left(z_{0,j} - \eta g_{1,j}^{\mathrm{sum}}, \eta \theta_1 \theta_0 \lambda_1\right). \end{aligned}$$

Next we consider the case $k \geq 2$. We show that

$$x_{k-1,j} = \frac{\theta_{k_j} \theta_{k_j-1}}{\theta_{k-1} \theta_{k-2}} x_{k_j,j} + \frac{1}{\theta_{k-1} \theta_{k-2}} \sum_{k'=k_j+2}^{k} \theta_{k'-2} z_{k'-1}. \tag{7}$$

For $k = k_j + 1$, (7) holds. Assume that (7) holds for some $k' \geq k_j + 1$. Then

$$
\begin{aligned}
x_{k',j} &= \left(1 - \frac{1}{\theta_{k'}}\right) x_{k'-1,j} + \frac{1}{\theta_{k'}} z_{k',j} \\
&= \left(1 - \frac{1}{\theta_{k'}}\right) \frac{\theta_{k_j} \theta_{k_j-1}}{\theta_{k'-1}\theta_{k'-2}} x_{k_j,j} + \left(1 - \frac{1}{\theta_{k'}}\right) \frac{1}{\theta_{k'-1}\theta_{k'-2}} \sum_{k''=k_j+2}^{k'} \theta_{k''-2} z_{k''-1} + \frac{1}{\theta_{k'}} z_{k',j} \\
&= \frac{\theta_{k_j} \theta_{k_j-1}}{\theta_{k'}\theta_{k'-1}} x_{k_j,j} + \frac{1}{\theta_{k'}\theta_{k'-1}} \sum_{k''=k_j+2}^{k'} \theta_{k''-2} z_{k''-1} + \frac{1}{\theta_{k'}} z_{k',j} \\
&= \frac{\theta_{k_j} \theta_{k_j-1}}{\theta_{k'}\theta_{k'-1}} x_{k_j,j} + \frac{1}{\theta_{k'}\theta_{k'-1}} \sum_{k''=k_j+2}^{k'+1} \theta_{k''-2} z_{k''-1}.
\end{aligned}
$$

The first equality is due to the definition of $x_{k'}$. The second equality follows from the assumption of induction. The third equality holds by Lemma A.1. This shows that (7) holds.

Next we show that

$$
z_{k'-1,j} = \frac{1}{1 + \eta\theta_{k'-1}\theta_{k'-2}\lambda_2} \text{soft}\left(z_{0,j} - \eta g_{k_j,j}^{\text{sum}} - \eta(\theta_{k'-1}\theta_{k'-2} - \theta_{k_j}\theta_{k_j-1})\widetilde{\nabla}_j, \eta\theta_{k'-1}\theta_{k'-2}\lambda_1\right),
$$
(8)

for $k' \in [k_j + 2, k]$.

By the definition of $z_{k'-1}$, we have that

$$
\begin{aligned}
z_{k'-1,j} &= \text{prox}_{\eta\theta_{k'-1}\theta_{k'-2}R}(z_0 - \eta\theta_{k'-1}\theta_{k'-2}\bar{g}_{k'-1})_j \\
&= \frac{1}{1 + \eta\theta_{k'-1}\theta_{k'-2}\lambda_2} \text{soft}(z_{0,j} - \eta\theta_{k'-1}\theta_{k'-2}\bar{g}_{k'-1,j}, \eta\theta_{k'-1}\theta_{k'-2}\lambda_1)
\end{aligned}
$$

From Lemma A.4, we see that

$$
\begin{aligned}
\theta_{k'-1}\theta_{k'-2}\bar{g}_{k'-1,j} &= \sum_{k''=1}^{k'-1} \theta_{k''-1} g_{k'',j} \\
&= \sum_{k''=1}^{k_j} \theta_{k''-1} g_{k'',j} + \left(\sum_{k''=k_j+1}^{k'-1} \theta_{k''-1}\right)\widetilde{\nabla}_j \\
&= g_{k_j,j}^{\text{sum}} + (\theta_{k'-1}\theta_{k'-2} - \theta_{k_j}\theta_{k_j-1})\widetilde{\nabla}_j.
\end{aligned}
$$

The first and third equality are due to Lemma A.2. The second equality holds because $g_{k''-1,j} = \widetilde{\nabla}_j$ for $k'' \in [k_j + 1, k - 1]$ by the assumption. This shows that (8) holds. Observe that

$$
\begin{aligned}
z_{k'-1,j} &= \frac{1}{1 + \eta\theta_{k'-1}\theta_{k'-2}\lambda_2} \text{soft}\left(z_{0,j} - \eta g_{k_j,j}^{\text{sum}} - \eta(\theta_{k'-1}\theta_{k'-2} - \theta_{k_j}\theta_{k_j-1})\widetilde{\nabla}_j, \eta\theta_{k'-1}\theta_{k'-2}\lambda_1\right) \\
&= \frac{1}{1 + \eta\theta_{k'-1}\theta_{k'-2}\lambda_2} \text{sign}\left(z_{0,j} - \eta g_{k_j,j}^{\text{sum}} - \eta(\theta_{k'-1}\theta_{k'-2} - \theta_{k_j}\theta_{k_j-1})\widetilde{\nabla}_j\right) \\
&\quad \times \max\left\{\left|z_{0,j} - \eta g_{k_j,j}^{\text{sum}} - \eta(\theta_{k'-1}\theta_{k'-2} - \theta_{k_j}\theta_{k_j-1})\widetilde{\nabla}_j\right| - \eta\theta_{k'-1}\theta_{k'-2}\lambda_1, 0\right\} \\
&= \begin{cases} \frac{1}{1+\eta\theta_{k'-1}\theta_{k'-2}\lambda_2}(z_{0,j} - M_{k',j}^+) & (z_{0,j} > M_{k',j}^+) \\ 0 & (M_{k',j}^- \leq z_{0,j} \leq M_{k',j}^+) \\ \frac{1}{1+\eta\theta_{k'-1}\theta_{k'-2}\lambda_2}(z_{0,j} - M_{k',j}^-) & (z_{0,j} < M_{k',j}^-) \end{cases},
\end{aligned}
$$

where $M_{k',j}^{\pm} = \eta\theta_{k'-1}\theta_{k'-2}(\widetilde{\nabla}_j \pm \lambda_1) + \eta g_{k_j,j}^{\text{sum}} - \eta\theta_{k_j}\theta_{k_j-1}\widetilde{\nabla}_j$. We define the real valued functions $M^{\pm}$ as follows:

$$
M_j^{\pm}(x) \stackrel{\text{def}}{=} (c_1 \pm c_2)x^2 - (c_1 \pm c_2)x + c_3,
$$

where $c_1 = \frac{\eta \widetilde{\nabla}_j}{4}$, $c_2 = \frac{\eta \lambda_1}{4}$ and $c_3 = \eta g_{k_j,j}^{\text{sum}} - \eta \theta_{k_j} \theta_{k_j-1} \widetilde{\nabla}_j$ Then we see that $M_j^{\pm}(k') = M_{k',j}^{\pm}$.
Let

$$D_{\pm} \overset{\text{def}}{=} (c_1 \pm c_2)^2 + 4(c_1 \pm c_2)(z_{0,j} - c_3),$$

$$s_{\pm}^+ \overset{\text{def}}{=} \frac{c_1 + c_2 \pm \sqrt{D_+}}{c_1 + c_2},$$

$$s_{\pm}^- \overset{\text{def}}{=} \frac{c_1 - c_2 \pm \sqrt{D_-}}{c_1 - c_2},$$

where if $s_{\pm}^{\pm}$ are not well defined, we simply assign 0 (or any number) to $s_{\pm}^{\pm}$. We can easily show that the following results:
1) If $c_1 > c_2$, then

$$z_{0,j} > M_j^+(x) \iff \begin{cases} x \in \emptyset & (D_+ \leq 0) \\ s_-^+ < x < s_+^+ & (D_+ > 0) \end{cases},$$

$$z_{0,j} < M_j^-(x) \iff \begin{cases} x \in \mathbb{R} & (D_- \leq 0) \\ x < s_-^- \vee x > s_+^- & (D_- > 0) \end{cases}.$$

2) If $c_1 = c_2$, thenãĂĂ

$$z_{0,j} > M_j^+(x) \iff \begin{cases} x \in \emptyset & (c_2 = 0 \wedge z_{0,j} \leq c_3) \\ x \in \mathbb{R} & (c_2 = 0 \wedge z_{0,j} > c_3) \\ x \in \emptyset & (c_2 > 0 \wedge D_+ \leq 0) \\ s_-^+ < x < s_+^+ & (c_2 > 0 \wedge D_+ > 0) \end{cases},$$

$$z_{0,j} < M_j^-(x) \iff \begin{cases} x \in \mathbb{R} & (z_{0,j} < c_3) \\ x \in \emptyset & (z_{0,j} \geq c_3) \end{cases}.$$

3) If $|c_1| < c_2$, then

$$z_{0,j} > M_j^+(x) \iff \begin{cases} x \in \emptyset & (D_+ \leq 0) \\ s_-^+ < x < s_+^+ & (D_+ > 0) \end{cases},$$

$$z_{0,j} < M_j^-(x) \iff \begin{cases} x \in \emptyset & (D_- \leq 0) \\ s_-^- < x < s_+^- & (D_- > 0) \end{cases}.$$

4) If $c_1 = -c_2$, then

$$z_{0,j} > M_j^+(x) \iff \begin{cases} x \in \emptyset & (z_{0,j} \leq c_3) \\ x \in \mathbb{R} & (z_{0,j} > c_3) \end{cases},$$

$$z_{0,j} < M_j^-(x) \iff \begin{cases} x \in \mathbb{R} & (c_2 = 0 \wedge z_{0,j} < c_3) \\ x \in \emptyset & (c_2 = 0 \wedge z_{0,j} \geq c_3) \\ x \in \emptyset & (c_2 > 0 \wedge D_- \leq 0) \\ s_-^- < x < s_+^- & (c_2 > 0 \wedge D_- > 0) \end{cases}.$$

5) If $c_1 < -c_2$, then

$$z_{0,j} > M_j^+(x) \iff \begin{cases} x \in \mathbb{R} & (D_+ \leq 0) \\ x < s_-^+ \vee x > s_+^+ & (D_+ > 0) \end{cases},$$

$$z_{0,j} < M_j^-(x) \iff \begin{cases} x \in \emptyset & (D_- \leq 0) \\ s_-^- < x < s_+^- & (D_- > 0) \end{cases}.$$

The lazy update rules of $x_{k-1,j}$ is derived by combining (7) with these results and noting that $k' \in [k_j + 2, k]$. Finally, combining the definition $y_{k,j} = (1 - 1/\theta_k)x_{k-1,j} + (1/\theta_k)z_{k-1,j}$ with (8) gives the lazy update of $y_{k,j}$. The update rule of $z_{k,j}$ is obvious from the proof of (8). $\qquad \square$

## F Experimental Details

In this section, we give the experimental details.

The details of the implemented algorithms and their parameter tunings were as follows:

For non-strongly convex cases ($(\lambda_1, \lambda_2) = (10^{-4}, 0)$),

- SVRG$^{++}$ [2] with default initial epoch length $m = n/(4b)$ [2]. We tuned only the learning rate.

- AccProxSVRG [8]. We tuned the epoch length, the constant momentum rate and the learning rate, and additional dummy $\ell_2$ regularizer weight for handling a non-strongly convex objective.

- UC [5] + SVRG [10] with default epoch length $m = 2n/b$ [3]. We tuned $\kappa$ in [5] and the learning rate. We fixed $\eta = 1$ in the algorithm of UC (note that $\eta$ is not learning rate).

- UC + AccProxSVRG. We tuned $\kappa$ in [5], the epoch length, the constant momentum rate and the learning rate. We fixed $\eta = 1$ in the algorithm of UC (note that $\eta$ is not learning rate).

- APCG [6]. We tuned the convexity parameter of the dual objective and the learning rate, and additional dummy $\ell_2$ regularizer weight for handling a non-strongly convex objective.

- Katyusha$^{ns}$ [1] with default epoch length $m = 2n/b$ and Katyusha momentum $\tau_2 = 1/2$ following the suggestion of [1]. We tuned only the learning rate. We did not adopt AdaptReg scheme because Katyusha with AdaptReg was always a bit slower than vanilla Katyusha in our experiments.

- DASVRDA$^{ns}$ with epoch length $m = n/b$ and $\gamma = \gamma_*$. We tuned only the learning rate.

- Adaptive Restart DASVRDA with epoch length $m = n/b$ and $\gamma = \gamma_*$. We tuned only the learning rate. We used the gradient scheme for the adaptive restarting, that is we restart DASVRDA$^{ns}$ if $(\widetilde{y}_s - \widetilde{x}_s)^\top (\widetilde{y}_{s+1} - \widetilde{x}_s) > 0$.

For strongly convex cases ($(\lambda_1, \lambda_2) = (10^{-4}, 10^{-6}), (0, 10^{-6})$),

- SVRG [10] with default epoch length $m = 2n/b$. We tuned only the learning rate.

- AccProxSVRG [8]. We tuned the epoch length, the constant momentum rate and the learning rate.

- UC [5] + SVRG [10] with default epoch length $m = 2n/b$ [4]. We tuned $\kappa$, $q$ in [5] and the learning rate.

- UC + AccProxSVRG. We tuned $\kappa$, $q$ in [5], the epoch length, the constant momentum rate and the learning rate.

- APCG [6]. We tuned the convexity parameter of the dual objective and the learning rate.

- Katyusha [1] with default epoch length $m = 2n/b$ and Katyusha momentum $\tau_2 = 1/2$ following the suggestion of [1]. We tuned $\tau_1$ in [1] and the learning rate.

- DASVRDA$^{sc}$ with epoch length $m = n/b$ and $\gamma = \gamma_*$. We tuned the fixed restart interval $S$ and the learning rate.

- Adaptive Restart DASVRDA with epoch length $m = n/b$ and $\gamma = \gamma_*$. We tuned only the learning rate. We use the gradient scheme for the adaptive restarting, that is we restart DASVRDA$^{ns}$ if $(\widetilde{y}_s - \widetilde{x}_s)^\top (\widetilde{y}_{s+1} - \widetilde{x}_s) > 0$.

For tuning the parameters, we chose the values that led to the minimum objective value. We selected the learning rates from the set $\{10^p, 2 \times 10^p, 5 \times 10^p \mid p \in \{0, \pm 1, \pm 2\}\}$ for each algorithm. We selected the epoch lengths from the set $\{n \times 10^{-k}, 2n \times 10^{-k}, 5n \times 10^{-k} \mid k \in \{0, 1, 2, 3\}\}$ and

the momentum rates from the set $\{1 - 10^{-k} \mid k \in \{1, 2, 3, 4\}\}$ for AccProxSVRG. We chose the additional dummy $\ell_2$ regularizer weights from the set $\{10^{-k}, 0 \mid k \in \{4, 5, 6, 8, 12\}\}$ for AccSVRG and APCG. We selected $\kappa, q$ from the set $\{10^{-k} \mid k \in \{1, 2, 3, 4, 5, 6\}\}$ for UC. We chose the convexity parameter from the set $\{10^{-k} \mid k \in \{3, 4, 5, 6, 7\}\}$ for APCG. We selected $\tau_1$ from the set $\{10^{-k}, 2 \times 10^{-k}, 5 \times 10^{-k} \mid k \in \{1, 2, 3\}\}$ for Katyusha. We selected the restart interval from the set $\{10^k, 2 \times 10^k, 5 \times 10^k \mid k \in \{0, 1, 2\}\}$ for DASVRDA$^{\text{sc}}$.

We fixed the initial points $0 \in \mathbb{R}^d$ for all algorithms.

For a fair comparison, we used uniform sampling for all algorithms, because AccProxSVRG does not support non-uniform sampling.

## G    DASVRG method

In this section, we briefly discuss a SVRG version of DASVRDA method (we call this algorithm DASVRG) and show that DASVRG has the same rates as DASVRDA.

In Section 4, we apply the double acceleration scheme to SVRDA method. We can also apply the one to SVRG. The only difference from DASVRDA is the update of $z_t$ in AccSVRDA (Algorithm 7). We take the following update for DASVRG:

$$z_k = \operatorname*{argmin}_{z \in \mathbb{R}^d} \left\{ \langle g_k, z \rangle + R(z) + \frac{1}{2\eta\theta_{k-1}} \|z - z_{k-1}\|^2 \right\} = \operatorname{prox}_{\eta\theta_{k-1}R} (z_{k-1} - \eta\theta_{k-1}g_k). \quad (9)$$

For the convergence analysis of DASVRG, we only need to show that Lemma A.5 is still valid for this algorithm.

*Proof of Lemma A.5 for DASVRG.* From (2) in the proof of Lemma A.5 for DASVRDA, we also have

$$\theta_k\theta_{k-1}P(x_k) \le \theta_{k-1}(\theta_k - 1)P(x_{k-1}) + \theta_{k-1}\hat{\ell}_k(z_k) + \frac{1}{2\eta}\|z_k - z_{k-1}\|^2$$
$$+ \frac{\theta_k\theta_{k-1}\|g_k - \nabla F(y_k)\|^2}{2\left(\frac{1}{\eta} - \bar{L}\right)} - \theta_{k-1}\langle g_k - \nabla F(y_k), z_{k-1} - y_k \rangle,$$

because the derivation of this inequality does not depend on the update rule of $z_t$. Observe that $z_k = \operatorname*{argmin}_{z \in \mathbb{R}^d} \{\theta_{k-1}\hat{\ell}_k(z) + 1/(2\eta)\|z - z_{k-1}\|^2\}$ from (9). Since $\theta_{k-1}\hat{\ell}_k(z) + 1/(2\eta)\|z - z_{k-1}\|^2$ is $\eta$-strongly convex, we have

$$\theta_{k-1}\hat{\ell}_k(z_k) + \frac{1}{2\eta}\|z_k - z_{k-1}\|^2 + \frac{1}{2\eta}\|z_k - x\|^2 \le \theta_{k-1}\hat{\ell}_k(x) + \frac{1}{2\eta}\|z_{k-1} - x\|^2.$$

Moreover, using the definitions of $\hat{\ell}$ and $\ell$, and Lemma A.3, we have

$$\hat{\ell}_k(x) = \ell_k(x) + \langle g_k - \nabla F(y_k), x - y_k \rangle$$
$$\le P(x) - \frac{1}{2\bar{L}}\frac{1}{n}\sum_{i=1}^n \frac{1}{nq_i}\|\nabla f_i(x) - \nabla f_i(y_k)\|^2 + \langle g_k - \nabla F(y_k), x - y_k \rangle.$$

Hence, we get

$$\theta_k\theta_{k-1}(P(x_k) - P(x)) \le \theta_{k-1}(\theta_k - 1)(P(x_{k-1}) - P(x)) + + \frac{1}{2\eta}(\|z_{k-1} - x\|^2 - \|z_k - x\|^2)$$
$$+ \frac{\theta_k\theta_{k-1}\|g_k - \nabla F(y_k)\|^2}{2\left(\frac{1}{\eta} - \bar{L}\right)} - \frac{\theta_{k-1}}{2\bar{L}}\frac{1}{n}\sum_{i=1}^n \frac{1}{nq_i}\|\nabla f_i(x) - \nabla f_i(y_k)\|^2$$
$$- \theta_{k-1}\langle g_k - \nabla F(y_k), z_{k-1} - x \rangle.$$

Note that $\theta_{k-1}(\theta_k - 1) \le \theta_{k-1}\theta_{k-2}$ for $k \ge 2$ and $\theta_1 = 1$. Finally, summing up the above inequality from $k = 1$ to $m$, dividing the both sides by $\theta_m\theta_{m-1}$ and taking expectations with respect to $I_k$ $(1 \le k \le m)$ give the desired result. $\qquad\square$

## Footnotes

[1]Note that we regard one computation of a full gradient as $n/b$ iterations in size $b$ mini-batch settings.

[2]In [2], the authors have suggested a default initial epoch length $m = n/4$. Since we used mini-batches with size $b$ in our experiments, it was natural to use $m = n/(4b)$. We made sure that using this epoch length improved the performances in all settings.

[3]In [10], the authors has suggested a default initial epoch length $m = 2n$. Since we used mini-batches with size $b$ in our experiments, it was natural to use $m = 2n/b$. We made sure that using this epoch length improved the performances in all settings.