[Reviews · NeurIPS 2017]

Reviewer 1



The paper proposes a doubly accelerated incremental method in mini-batch setting called DASVRDA, where on one hand it accelerates in the sense of Nesterov and on the other hand it accelerates by achieving a better dependency on the size of mini-batches. Convergence result has been provided for both strongly convex and convex objectives. I find the result very interesting, it is nice to see an acceleration occurs respect to the size of mini-batches both in theory and in the experiment. One important step in the algorithm is the restarting scheme which it is stated only require the optimal strong convexity of the objective instead of the ordinary strong convexity, can authors provide more details on it because I am confused by the terminology. In the experiment, an algorithm applying the UC[8](universal catalyst) on AccProxSVRG[12] (the accelerated mini-batch proximal stochastic variance reduce gradient) has been implemented but its theoretical guarantee is not mentioned in the paper. However, this method may be the real competitor to be compared with. Because it also uses an inner-outer loop construction as DASVRDA and the resulting algorithm will potentially enjoy the double acceleration property. Moreover, it can be applied on both strongly convex and convex settings. Can authors comment on it? Overall, the paper provides interesting result but some details need to be clarified. #EDIT Authors' feedback have addressed and answered my concerns.

Reviewer 2



Doubly Accelerated Stochastic Variance Reduced Dual Averaging Method for Regularized Empirical Risk Minimization This paper introduce a new accelerated version of SVRG algorithm with mini batching. They consider two type of objective functions: strongly convex and non strongly convex functions. Their assumptions for both settings are fairly weak and for strongly convex functions, they assume weaker condition than the vanilla SVRG algorithm. They add the momentum terms in inner loop and outer loop of their algorithm. Their proves show that their methods improves the current state of the art methods. They test their methods against some real world data sets and show empirically their methods outperform other methods in mini_batch settings. Positive Points: Using mini_batch in stochastic optimization of ML models could help with reducing variance and also it is parallelizable. They considered mini_batch for accelerated SVRG for convex and non convex function and improved the convergence bounds in this setting. They showed that to get optimal cost, the good option for mini batch size is square root of n. They computes the efficient value for all the parameter of the algorithms and make the implementation of the algorithm easy. The paper's writing is clear and easy to follow which making the reading of the paper enjoyable. Minor comments: -The only part that I think could be improved is the experiment section. I would rather to see the result of method on bigger data set with bigger than 1M samples with different mini batch size. Beside comparing test error is also useful for ML community. - When you refer to supp. mat. in main paper, it'd be better if you mention the section of supp. mat. in main paper. It makes reaching the related part in supp. faster.

Reviewer 3



The paper proposes a novel doubly accelerated variance reduced dual averaging method for solving the convex regularized empirical risk minimization problem in mini batch settings. The method essentially can be interpreted as replacing the proximal gradient update of APG method with the inner SVRG loop and then introducing momentum updates in inner SVRG loops. Finally to allow lazy updated, primal SVRG is replaced with variance reduce dual averaging. The main difference from AccProxSVRG is the introduction of momentum term at the outer iteration level also. The method requires only O(sqrt{n}) sized mini batches to achieve optimal iteration complexities for both convex and non-convex functions when the problem is badly conditioned or require high accuracy. Experimental results show substantial improvements over state of the art under the above scenario. Overall, given the theoretical complexity of the paper, it is very well written and explained. Relation with the previous work and differences are clear and elaborative. The reviewer thinks that the paper makes substantial theoretical and algorithmic advances. However, I am giving paper relatively lower rating due to following reasons: a) I would like to see more experiments on different datasets especially the effect of conditioning and regularizer. b) The accuracy curves are not shown in the experiments. This is important in this case because it seems that major differences between methods are visible only at lower values of the gap and it would be interesting to see if accuracy has saturated by then which will make the practical utility limited. c) Instead of x axis as gradient evaluations, I would like to see the actual wall clock time there because the constants or O(1) computation in inner loop matters a lot. Especially, the double accelerated equations look computationally more tedious and hence I would like to see the time axis there. d) Looking at lazy updates in supplementary, I am not sure why accelerated SVRG cannot have lazy updates. Can authors exactly point out the step in detail which will have the issues.